EMBO
Molecular Medicine

# FGF21 gene therapy as treatment for obesity and insulin resistance

Veronica Jimenez[1,2,3,†], Claudia Jambrina[1,2,3,†], Estefania Casana[1,2,3], Victor Sacristan[1,2,3], Sergio Muñoz[1,2,3], Sara Darriba[1,2,3], Jordi Rodó[1,2,3], Cristina Mallol[1,2,3], Miquel Garcia[1,2,3], Xavier León[1,2,3], Sara Marcó[1,2,3], Albert Ribera[1,2,3], Ivet Elias[1,2,3], Alba Casellas[1,2,3], Ignasi Grass[1,2,3], Gemma Elias[1,2,3], Tura Ferré[1,3], Sandra Motas[1,2,3], Sylvie Franckhauser[1,3], Francisca Mulero[3,4], Marc Navarro[1,3,5], Virginia Haurigot[1,2,3], Jesus Ruberte[1,3,5] & Fatima Bosch[1,2,3,*] ⓘD

## Abstract

Prevalence of type 2 diabetes (T2D) and obesity is increasing worldwide. Currently available therapies are not suited for all patients in the heterogeneous obese/T2D population, hence the need for novel treatments. Fibroblast growth factor 21 (FGF21) is considered a promising therapeutic agent for T2D/obesity. Native FGF21 has, however, poor pharmacokinetic properties, making gene therapy an attractive strategy to achieve sustained circulating levels of this protein. Here, adeno-associated viral vectors (AAV) were used to genetically engineer liver, adipose tissue, or skeletal muscle to secrete FGF21. Treatment of animals under long-term high-fat diet feeding or of ob/ob mice resulted in marked reductions in body weight, adipose tissue hypertrophy and inflammation, hepatic steatosis, inflammation and fibrosis, and insulin resistance for > 1 year. This therapeutic effect was achieved in the absence of side effects despite continuously elevated serum FGF21. Furthermore, FGF21 overproduction in healthy animals fed a standard diet prevented the increase in weight and insulin resistance associated with aging. Our study underscores the potential of FGF21 gene therapy to treat obesity, insulin resistance, and T2D.

**Keywords** AAV gene therapy; FGF21; insulin resistance; obesity; type 2 diabetes
**Subject Categories** Genetics, Gene Therapy & Genetic Disease; Metabolism

See also: **CH Sponton & S Kajimura** (August 2018)

## Introduction

The prevalence of type 2 diabetes (T2D) is growing at an alarming rate, and T2D has become a major health problem worldwide. T2D and insulin resistance are very strongly associated with obesity, whose prevalence is also increasing (So & Leung, 2016). Obesity increases the risk of mortality (Peeters et al, 2003) and is also a very significant risk factor for heart disease, immune dysfunction, hypertension, arthritis, neurodegenerative diseases, and certain types of cancer (Spiegelman & Hotamisligil, 1993; Whitmer, 2007; Roberts et al, 2010). Lifestyle intervention and conventional pharmacologic treatments have proven effective for many obese and T2D patients. However, these therapeutic options are not successful in all cases and are not exempt of undesirable side effects. Hence, there is a medical need for novel, well-tolerated treatments for the large and heterogenous population of obese/T2D patients.

Fibroblast growth factor 21 (FGF21) has recently emerged as a promising therapeutic agent for the treatment of obesity and T2D (Kharitonenkov & DiMarchi, 2017). This peptide hormone is secreted by several organs and can act on multiple tissues to regulate energy homeostasis (Potthoff et al, 2012; Fisher & Maratos-Flier, 2016). One of the main secretors of FGF21 is the liver (Markan et al, 2014). FGF21 binds specifically to the FGF receptors (FGFR) and needs β-Klotho as an obligate co-receptor (Fisher & Maratos-Flier, 2016). The administration of recombinant FGF21 protein to ob/ob, db/db, or high-fat diet (HFD)-fed mice or to obese Zucker diabetic fatty (ZDF) rats promotes a robust reduction in adiposity, lowers blood glucose and triglycerides, and improves insulin sensitivity (Kharitonenkov et al, 2005; Coskun et al, 2008; Berglund et al, 2009; Xu et al, 2009a; Adams et al, 2012a,b). Similarly, liver-specific overexpression of FGF21 in transgenic mice protects animals from diet-induced obesity and insulin resistance (Kharitonenkov et al, 2005; Inagaki et al, 2007). Moreover, the

1   Center of Animal Biotechnology and Gene Therapy (CBATEG), Universitat Autònoma de Barcelona, Bellaterra, Spain
2   Department of Biochemistry and Molecular Biology, Universitat Autònoma de Barcelona, Bellaterra, Spain
3   CIBER de Diabetes y Enfermedades Metabólicas Asociadas (CIBERDEM), Madrid, Spain
4   Molecular Imaging Unit, Spanish National Cancer Research Centre (CNIO), Madrid, Spain
5   Department of Animal Health and Anatomy, School of Veterinary Medicine, Universitat Autònoma de Barcelona, Bellaterra, Spain
    *Corresponding author. Tel: +34 93 581 41 82; E-mail: fatima.bosch@uab.es
    †These authors contributed equally to this work

administration of FGF21 to obese diabetic rhesus monkeys markedly reduces fasting plasma glucose, triglyceride, and insulin levels and induces a small but significant loss of body weight (Kharitonenkov et al, 2007).

The native FGF21 protein has, however, poor pharmacokinetic properties, including a short half-life (ranging from 0.5 to 2 h), in part because of increased glomerular filtration in the kidney due to its small size (~22 kDa), and susceptibility to in vivo proteolytic degradation and in vitro aggregation (Zhang & Li, 2015; So & Leung, 2016). The pharmaceutical industry is devoting considerable efforts to overcoming these limitations and improving the yield of production of FGF21 analogues or mimetics to enable the development of potential drug products (Zhang & Li, 2015; So & Leung, 2016). These FGF21-class molecules have been reported to have similar therapeutic efficacy than the native FGF21 protein in small and large animal models of obesity and T2D (Foltz et al, 2012; Hecht et al, 2012; Adams et al, 2013; Talukdar et al, 2016; Stanislaus et al, 2017). Indeed, first-generation FGF21 analogues have already reached the clinical stage, and reports from two phase I clinical trials have shown significant improvement of dyslipidemia, slight body weight loss, and reductions in fasting insulinemia in patients with obesity and T2D (Gaich et al, 2013; Talukdar et al, 2016). Thus, preclinical evidence and clinical evidence corroborate that FGF21 may be an attractive candidate to combat obesity and TD2.

Despite the pharmacokinetic improvements, FGF21 analogues/mimetics require periodic administrations to mediate clinical benefit, which may not only be uncomfortable for patients and compromise treatment compliance but may also raise immunological issues associated with the administration of exogenous proteins (Gaich et al, 2013; Talukdar et al, 2016; Kim et al, 2017). Here, we took advantage of adeno-associated viral (AAV) vectors and their ability to mediate multi-year production of therapeutic proteins (Mingozzi & High, 2011; Naldini, 2015) to develop a gene therapy strategy for obesity and T2D based on FGF21 gene transfer to the liver, adipose tissue, or skeletal muscle. AAV vectors, derived from non-pathogenic viruses, are predominantly non-integrative vectors that persist for years as episomes in the nucleus of non-dividing cells (Lisowski et al, 2015; Wolf et al, 2015). Multiple studies have reported high transduction efficiencies for in vivo gene transfer with these vectors, as well as excellent safety profiles in clinical studies (Mingozzi & High, 2011; Naldini, 2015). In particular, studies in large animal models (Rivera et al, 2005; Niemeyer et al, 2009; Callejas et al, 2013; Bainbridge et al, 2015), as well as clinical studies (Hauswirth et al, 2008; Maguire et al, 2008; Simonelli et al, 2010; Buchlis et al, 2012; Jacobson et al, 2012; Gaudet et al, 2013; Testa et al, 2013; Nathwani et al, 2014; Bainbridge et al, 2015), have provided strong evidence of AAV-mediated long-term expression for a variety of therapeutic proteins in the absence of clinically significant adverse events. Thus, AAV-mediated gene therapy can potentially overcome the pharmacokinetic limitations of FGF21-class molecules.

Here, we demonstrate for the first time that a single administration of AAV vectors encoding FGF21 enabled a long-lasting increase in FGF21 levels in circulation, which resulted in sustained counteraction of obesity, hepatic steatosis, and insulin resistance in two different models of obesity and T2D, the HFD-fed mouse and the ob/ob mouse. In healthy animals, it prevented age-associated weight gain and insulin resistance. Our results underscore the potential of FGF21 gene therapy to treat these conditions.

## Results

### Persistent reversion of obesity by liver-specific AAV8-mediated FGF21 overexpression

Two-month-old C57Bl6 mice ("young adults") were fed either a chow or a HFD for 10 weeks. During these first 2.5 months of follow-up, while the weight of chow-fed animals increased by 27%, animals fed a HFD became obese (72% body weight gain) (Fig 1A). Obese animals were then administered intravenously (IV) with $1 \times 10^{10}$ or $5 \times 10^{10}$ viral genomes (vg) of serotype 8 AAV vectors encoding a murine optimized FGF21 coding sequence, whose expression was under the control of the synthetic liver-specific hAAT promoter (AAV8-hAAT-FGF21). As controls, another cohort of obese mice and the cohort of chow-fed mice received $5 \times 10^{10}$ vg of non-coding null vectors (AAV8-null). Following AAV delivery, mice were maintained on chow or HFD feeding for about 1 year; that is, up to 16.5 months of age, and body weight and metabolic parameters were monitored regularly. Animals treated with $1 \times 10^{10}$ vg AAV8-hAAT-FGF21 gained as much weight as AAV8-null-injected HFD-fed mice (Fig 1A). In marked contrast, the cohort of mice treated with $5 \times 10^{10}$ vg AAV8-hAAT-FGF21 normalized their body weight within a few weeks of AAV delivery (Fig 1A). Indeed, the mean body weight of this group of animals became indistinguishable from that of the chow-fed, AAV8-null-injected cohort for the duration of the follow-up period (~1 year) (Fig 1A and Appendix Fig S1A).

Having observed the profound effects on body weight exerted by FGF21 gene transfer to the liver, and taking into account that obesity and T2D are pathologies associated with aging, we set out another study in which animals began HFD feeding when older, at

---

**Figure 1. AAV8-mediated liver gene transfer of FGF21 counteracts HFD-induced obesity.**

A, B  Evolution of body weight in animals treated with AAV8-hAAT-FGF21 as young adults (A) or as adults (B). C57Bl6 mice were fed a HFD for ~10 weeks and then administered with $1 \times 10^{10}$, $2 \times 10^{10}$, or $5 \times 10^{10}$ vg/mouse of AAV8-hAAT-FGF21 vectors. Control obese mice and control chow-fed mice received $5 \times 10^{10}$ vg of AAV8-hAAT-null.

C  Weight of the epididymal (eWAT), inguinal (iWAT), and retroperitoneal (rWAT) white adipose tissue depots, the liver, and the quadriceps obtained from mice treated with AAV8-hAAT-FGF21 vectors as young adults (top panel) or as adults (bottom panel).

D  Circulating levels of FGF21 at different time points after vector administration.

Data information: All values are expressed as mean ± SEM. In (A–D), young adults: AAV8-hAAT-null chow (n = 10 animals), AAV8-hAAT-null HFD (n = 8), AAV8-hAAT-FGF21 HFD $1 \times 10^{10}$ vg (n = 9), and $5 \times 10^{10}$ vg (n = 8). Adults: AAV8-hAAT-null chow (n = 7), AAV8-hAAT-null HFD (n = 7), AAV8-hAAT-FGF21 HFD $1 \times 10^{10}$ vg (n = 7), $2 \times 10^{10}$ vg (n = 8) and $5 \times 10^{10}$ vg (n = 7). In (A–D), data were analyzed by one-way ANOVA with Tukey's post hoc correction. *$P < 0.05$, **$P < 0.01$, and ***$P < 0.001$ versus the chow-fed null-injected group. #$P < 0.05$, ##$P < 0.01$, and ###$P < 0.001$ versus the HFD-fed null-injected group. HFD, high-fat diet.

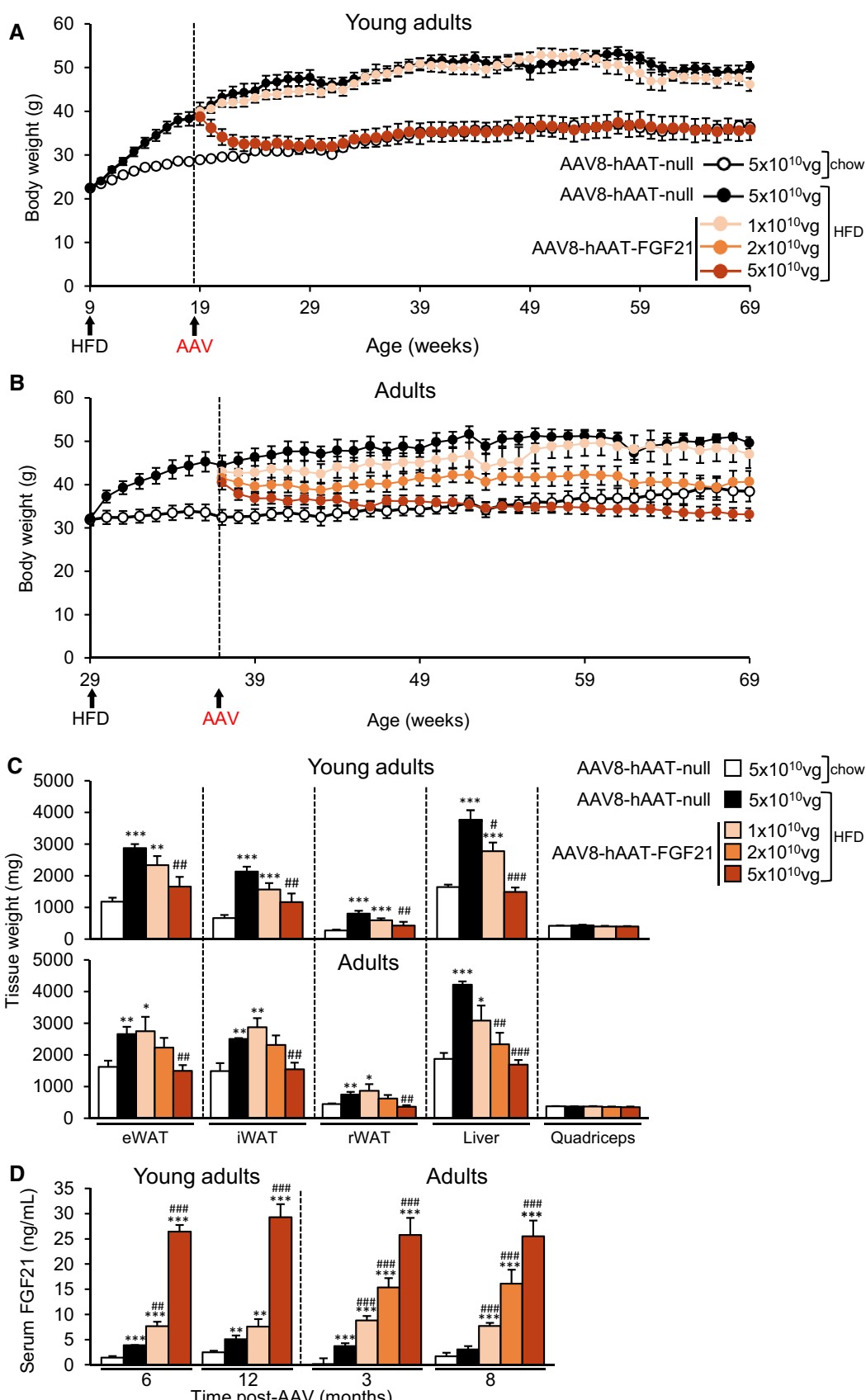

**Figure 1.**

the age of 29 weeks ("adults"). By week 38, older HFD-fed mice had also become obese, with a mean body weight gain of ~40% (Fig 1B). Mice were then administered IV with AAV8-hAAT-FGF21 vectors at three different doses: $1 \times 10^{10}$, $2 \times 10^{10}$, and $5 \times 10^{10}$ vg/mouse. As previously, chow and HFD-fed cohorts injected with AAV8-hAAT-null vectors ($5 \times 10^{10}$ vg) were used as controls. A clear dose-dependent loss of body weight was observed in the groups treated with AAV8-hAAT-FGF21. Similar to the observations made in younger mice, the lowest dose of vector did not counteract the weight gain associated with HFD feeding, although the mean weight of these animals was always lower than that of AAV8-null-injected HFD-fed mice (Fig 1B). Animals treated with $2 \times 10^{10}$ vg AAV8-hAAT-FGF21 initially lost 10% of body weight and continued to progressively lose weight to reach the weight of chow-fed animals toward the end of the study (~8 months) (Fig 1B and Appendix Fig S1B). Upon administration of $5 \times 10^{10}$ vg AAV8-hAAT-FGF21, adult mice also experienced progressive loss of body weight, which by the end of the study (16.5 months of age) was similar to the weight documented before the initiation of the HFD feeding (Fig 1B and Appendix Fig S1B).

The counteraction of obesity by AAV8-hAAT-FGF21 treatment was parallel to a dose-dependent decrease in the weight of the main white adipose tissue (WAT) depots, such as the epididymal (eWAT), inguinal (iWAT), and retroperitoneal (rWAT) fat pads, both in animals treated as young adults and in those treated as adults (Fig 1C and Appendix Fig S1C). The HFD-induced increase in the weight of the liver was completely normalized by FGF21 gene transfer at the highest doses of vector used, whereas the weight of the quadriceps was unchanged by the diet or AAV delivery (Fig 1C and Appendix Fig S1D).

AAV8-hAAT-FGF21-treated mice of both ages showed specific overexpression of codon-optimized FGF21 in the liver (Appendix Fig S1E), which resulted in secretion of FGF21 into the bloodstream in a dose-dependent manner in both groups of mice, with levels remaining stable for up to 1 year after a single administration of the vector (Fig 1D).

**Liver overproduction of FGF21 reverses HFD-associated WAT hypertrophy and inflammation**

HFD feeding induces an increase in the size of WAT adipocytes (Sattar & Gill, 2014). Administration of FGF21-encoding vectors

counteracted this increase (Fig 2A). Morphometric analysis of WAT revealed that the area of white adipocytes of animals treated as young adults with $1 \times 10^{10}$ or $5 \times 10^{10}$ vg of vector and of mice treated as adults with $2 \times 10^{10}$ or $5 \times 10^{10}$ vg of vector was similar to that of animals fed a chow diet (Fig 2B). In both groups of FGF21-treated animals, there was a redistribution of the size of adipocytes, with a greater proportion of smaller adipocytes (Appendix Fig S2A). In agreement with the decrease in adiposity and reversal of WAT hypertrophy, adiponectin and leptin levels were also normalized in animals treated with highest doses of AAV8-hAAT-FGF21 vectors, irrespective of the age of initiation of the treatment (Fig 2C and D).

Obesity also causes the inflammation of WAT (Hajer et al, 2008). Thus, we analyzed inflammation in this tissue through immunostaining for the macrophage-specific marker Mac2 and the expression of pro-inflammatory molecules. While HFD-fed mice showed increased presence of macrophages, revealed as "crown-like" structures, in the eWAT, animals treated as young adults or adults with $5 \times 10^{10}$ vg AAV8-hAAT-FGF21 had no sign of macrophage infiltration (Fig 2E and Appendix Fig S2B). This was parallel to the normalization in the expression of the macrophage markers F480 and CD68 and of the pro-inflammatory cytokines TNF-α and IL-1β (Fig 2F–H and Appendix Fig S2C–E), indicating that FGF21 expression counteracted the inflammation of WAT associated with obesity.

**FGF21 overproduction by the liver enhances energy expenditure without inducing browning of subcutaneous WAT**

The loss of body weight observed in the animals treated with AAV8-hAAT-FGF21 vectors was not due to a reduction in food intake. Weekly measurements of the number of calories consumed per day revealed that FGF21-treated animals actually ate more than controls groups (Fig 3A). When indirect calorimetry was performed 2 months after AAV delivery, both mice treated as young adults and those treated as adults with $5 \times 10^{10}$ vg AAV8-hAAT-FGF21 and fed a HFD had much higher energy expenditure during the light and dark cycles than age-matched control groups (Fig 3B). Similar results were obtained in animals treated as young adults 10 months after AAV8-hAAT-FGF21 delivery (Appendix Fig S3A).

This observation was in agreement with AAV8-hAAT-FGF21-mediated effects on locomotor activity. In contrast to the hypoactivity observed in the open-field test in the animals fed a HFD that

**Figure 2.   Reversal of WAT hypertrophy and inflammation by AAV8-hAAT-FGF21 treatment.**

A       Representative images of the hematoxylin–eosin staining of the eWAT from animals fed a chow or a HFD and administered with either AAV8-hAAT-null or $5 \times 10^{10}$ vg/mouse AAV8-hAAT-FGF21 vectors as young adults (*left panels*) or adults (*right panels*). While HFD-fed, null-injected mice had larger adipocytes, HFD-fed, FGF21-treated animals had adipocytes of reduced size. Scale bars: 100 μm.

B       Morphometric analysis of the area of WAT adipocytes in animals treated as young adults or as adults.

C, D    Circulating levels of adiponectin (C) and leptin (D).

E       Immunohistochemistry for the macrophage-specific marker Mac2 in eWAT sections from animals that received $5 \times 10^{10}$ vg/mouse AAV8-hAAT-FGF21 as adults. The micrographs illustrate the presence of crown-like structures (*red arrows* and inset) in the eWAT of HFD-fed, null-injected animals but not in the eWAT of HFD-fed, FGF21-treated mice. Scale bars: 200 μm and 50 μm (inset).

F–H     Quantification by qRT–PCR of the expression of the markers of inflammation F4/80 (F), IL-1β (G), and TNF-α (H) in the group of animals that initiated the HFD feeding and received FGF21 vectors as adults.

Data information: All values are expressed as mean ± SEM. In (B), $n = 4$ animals/group. In (C, D, F–H), young adults: AAV8-hAAT-null chow ($n = 10$ animals), AAV8-hAAT-null HFD ($n = 8$), AAV8-hAAT-FGF21 HFD $1 \times 10^{10}$ vg ($n = 9$), and $5 \times 10^{10}$ vg ($n = 8$). Adults: AAV8-hAAT-null chow ($n = 7$), AAV8-hAAT-null HFD ($n = 7$), AAV8-hAAT-FGF21 HFD $1 \times 10^{10}$ vg ($n = 7$), $2 \times 10^{10}$ vg ($n = 8$), and $5 \times 10^{10}$ vg ($n = 7$). In (B–D, F–H), data were analyzed by one-way ANOVA with Tukey's post hoc correction. *$P < 0.05$, **$P < 0.01$, and ***$P < 0.001$ versus the chow-fed null-injected group. #$P < 0.05$, ##$P < 0.01$, and ###$P < 0.001$ versus the HFD-fed null-injected group. HFD, high-fat diet.

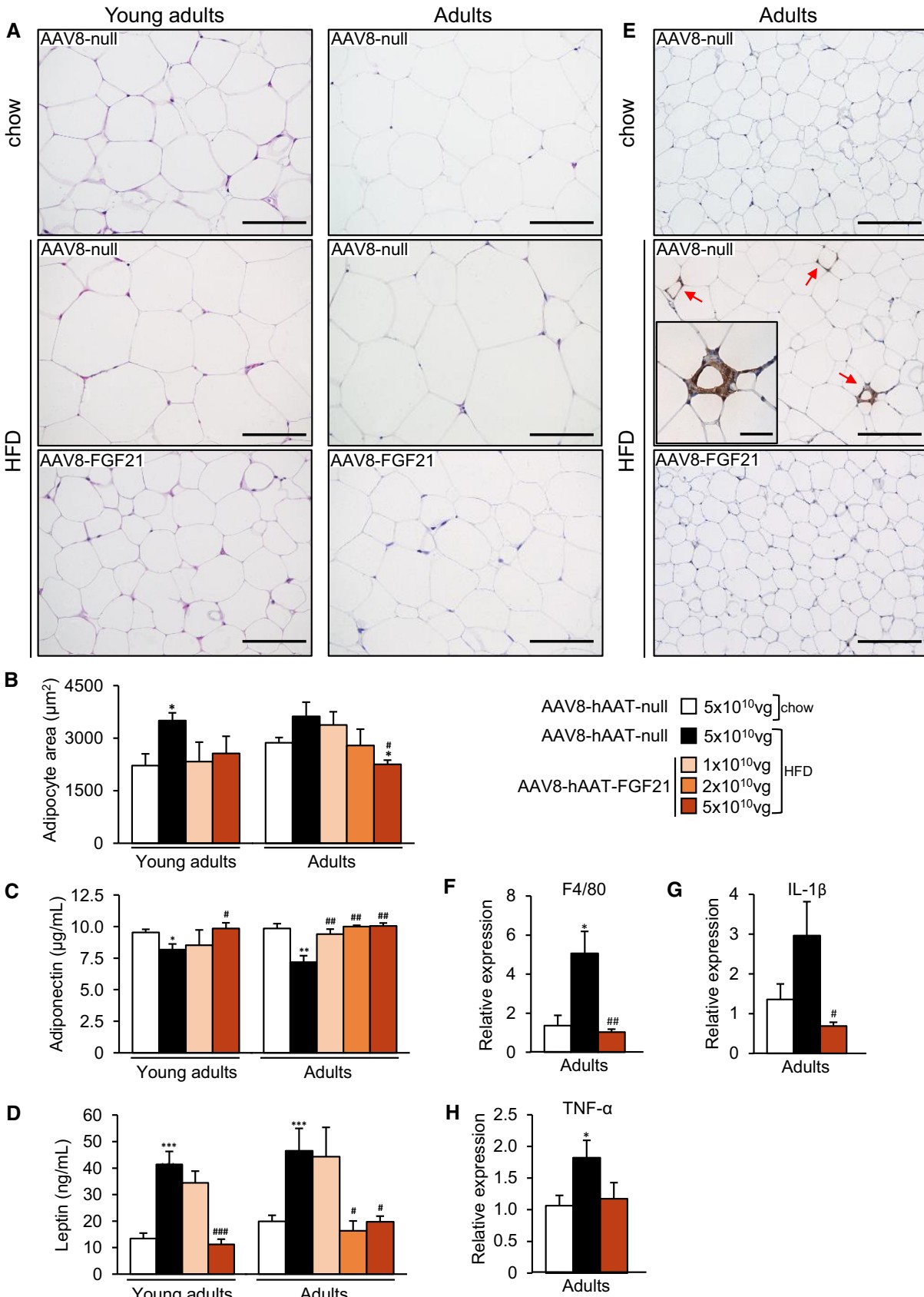

**Figure 2.**

received AAV8-null vectors, mice treated with $5 \times 10^{10}$ vg AAV8-hAAT-FGF21 as young adults showed the same degree of spontaneous locomotor activity than chow-fed, null-injected animals. As shown in Fig 3C, 1 year after AAV8-hAAT-FGF21 delivery, treated animals travelled more distance, rested less time, and spent more time doing slow and fast movements than untreated HFD-fed controls.

Given that changes in energy expenditure may reflect changes in thermogenesis, we evaluated the degree of activation of the brown adipose tissue (BAT). Both mice treated as young adults and those treated as adults with $5 \times 10^{10}$ vg AAV8-hAAT-FGF21 showed decreased lipid deposition in BAT (Fig 3D). The content of UCP1 protein in BAT was increased in a dose-dependent manner in mice treated with AAV8-hAAT-FGF21 vectors as young adults (Fig 3E), consistent with an increase in non-shivering thermogenesis induced by FGF21 gene transfer to the liver.

The browning of the subcutaneous WAT, characterized by the appearance of beige adipocytes, is also associated with increases in energy expenditure (Harms & Seale, 2013). To evaluate whether browning was accountable for the enhancement of energy expenditure observed following AAV8-hAAT-FGF21 treatment, histological evaluation of iWAT was performed. In agreement with the decreased weight of this pad (Fig 1C), the adipocytes of HFD-fed AAV8-hAAT-FGF21-treated animals were smaller than those of HFD-fed null-injected animals (Fig 3F). Treatment with AAV8-hAAT-FGF21 vectors, nevertheless, did not result in increased detection of multilocular beige adipocytes in iWAT at any of the doses tested, either in animals treated as young adults or in those treated as adults (Fig 3F). Accordingly, there were no statistically significant differences in the levels of UCP1 protein in iWAT between the HFD-fed groups (Appendix Fig S3B).

The creatine-driven substrate cycle and sarco/endoplasmic reticulum $Ca^{2+}$-ATPase 2b (Serca2b)-mediated calcium cycling can increase thermogenesis in iWAT independently of UCP1 (Kazak et al, 2015; Ikeda et al, 2017). Higher levels of expression of phosphatase orphan 1 (Phospho1), an enzyme involved in the creatine-driven substrate cycle, were observed in iWAT of HFD-fed mice treated with $5 \times 10^{10}$ vg of AAV8-hAAT-FGF21 when compared with age-matched, chow- and HFD-fed control groups (Fig 3G),

suggesting that the activity of the creatine-driven cycle was probably increased as a result of FGF21 gene transfer. Regarding the calcium cycling-dependent thermogenic mechanism, no differences in the expression levels of Serca2b were detected in the iWAT of animals treated with AAV8-hAAT-FGF21 vectors when compared with chow- or HFD-fed null-treated animals (Appendix Fig S3C). On the other hand, the iWAT expression of ryanodine receptor 2 (RyR2), another enzyme involved in the same cycle, was increased by HFD feeding in both null- and AAV8-hAAT-FGF21-treated mice (Appendix Fig S3C). Altogether, these results suggest that the calcium cycling-dependent thermogenic mechanism is not involved in the improvement of whole-body energy homeostasis observed after AAV-FGF21 treatment.

## Reversal of hepatic steatosis, inflammation, and fibrosis by liver FGF21 overproduction

Histological analysis of the liver showed that all null-treated animals fed a HFD had marked hepatic steatosis at the time of sacrifice (Fig 4A–D). In contrast, HFD-fed mice receiving $5 \times 10^{10}$ vg AAV8-hAAT-FGF21 as young adults or as adults evidenced reversal of this pathological deposition of lipids (Fig 4A). These histological findings were parallel to a marked reduction in the total liver triglyceride and cholesterol content of $5 \times 10^{10}$ vg AAV8-hAAT-FGF21-treated animals (Fig 4B and C). In addition, animals fed a HFD and treated with $5 \times 10^{10}$ vg AAV8-hAAT-FGF21 vectors when young adults or adults showed no sign of hepatic inflammation, as evidenced by the lack of staining for Mac2, which revealed increased presence of macrophages in the livers of null-treated HFD-fed mice (Fig 4D). Finally, FGF21 gene transfer to the liver reversed hepatic fibrosis. While collagen fibers were readily detectable following PicroSirius Red staining or Masson's trichrome staining of liver sections from animals fed a HFD and injected with control null vectors, they were undetectable in the livers of AAV8-hAAT-FGF21-treated mice (Fig EV1 and Appendix Fig S4). These mice also showed markedly reduced hepatic expression of collagen 1 (Fig 4E and F). Altogether, these findings indicated that AAV8-hAAT-FGF21 treatment protected from the development of HFD-induced non-alcoholic steatohepatitis (NASH).

---

**Figure 3.    AAV8-hAAT-FGF21-mediated increased energy expenditure and decreased fat accumulation in iBAT and iWAT.**

A    Histogram depicting the food intake of animals fed a chow or a HFD and administered with either AAV8-hAAT-null or AAV8-hAAT-FGF21 vectors as young adults or adults.

B    Energy expenditure was measured with an indirect open circuit calorimeter in all experimental groups 2 months after AAV8-hAAT-null or AAV8-hAAT-FGF21 vector delivery. Data were taken during the light and dark cycles.

C    Assessment of the locomotor activity through the open-field test in animals that had been subjected to HFD feeding since ~2 months of age and were treated with either null or FGF21-encoding vectors 2 months later (young adults).

D    Hematoxylin–eosin staining of BAT tissue sections obtained from the same cohort of animals as in (A). Scale bars: 50 μm.

E    Western blot analysis of UCP1 content in BAT from the same cohort of animals as in (C). A representative immunoblot is shown (top). The histogram depicts the densitometric analysis of two different immunoblots (bottom).

F    Hematoxylin–eosin staining of iWAT tissue sections obtained from the same cohort of animals as in (A). Scale bars: 100 μm.

G    Quantification by qRT–PCR of the expression of Phospho1 in iWAT in the groups of animals that initiated the HFD feeding and received FGF21 vectors as young adults or adults.

Data information: All values are expressed as mean ± SEM. In (A–C, G), young adults: AAV8-hAAT-null chow ($n = 10$ animals), AAV8-hAAT-null HFD ($n = 8$), AAV8-hAAT-FGF21 HFD $1 \times 10^{10}$ vg ($n = 9$), and $5 \times 10^{10}$ vg ($n = 8$). Adults: AAV8-hAAT-null chow ($n = 7$), AAV8-hAAT-null HFD ($n = 7$), AAV8-hAAT-FGF21 HFD $1 \times 10^{10}$ vg ($n = 7$), $2 \times 10^{10}$ vg ($n = 8$), and $5 \times 10^{10}$ vg ($n = 7$). In (E), $n = 4$ animals/group. In (A, E, G), data were analyzed by one-way ANOVA with Tukey's post hoc correction.
*$P < 0.05$, **$P < 0.01$, and ***$P < 0.001$ versus the chow-fed null-injected group. #$P < 0.05$ and ###$P < 0.001$ versus the HFD-fed null-injected group. HFD, high-fat diet.
Source data are available online for this figure.

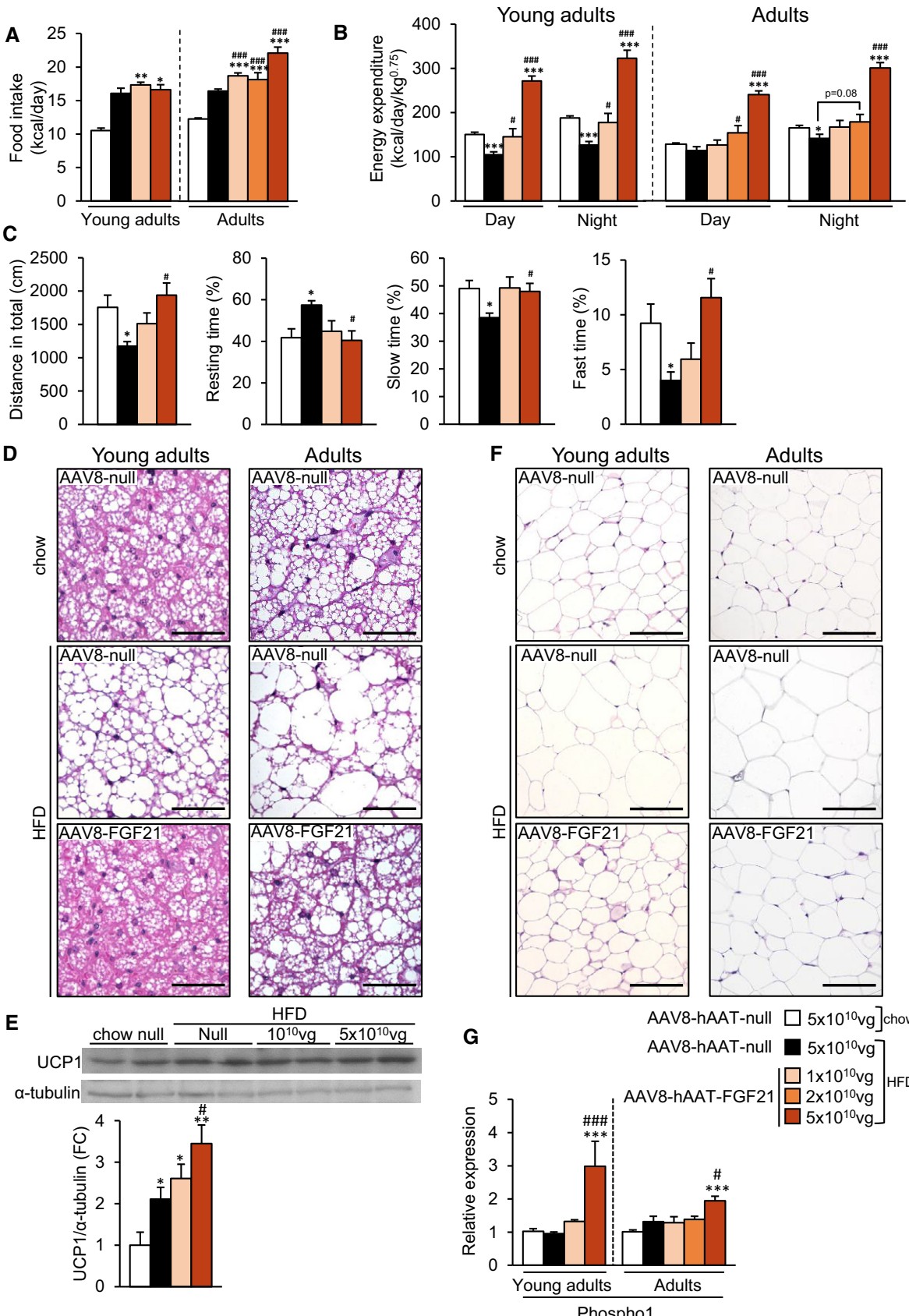

**Figure 3.**

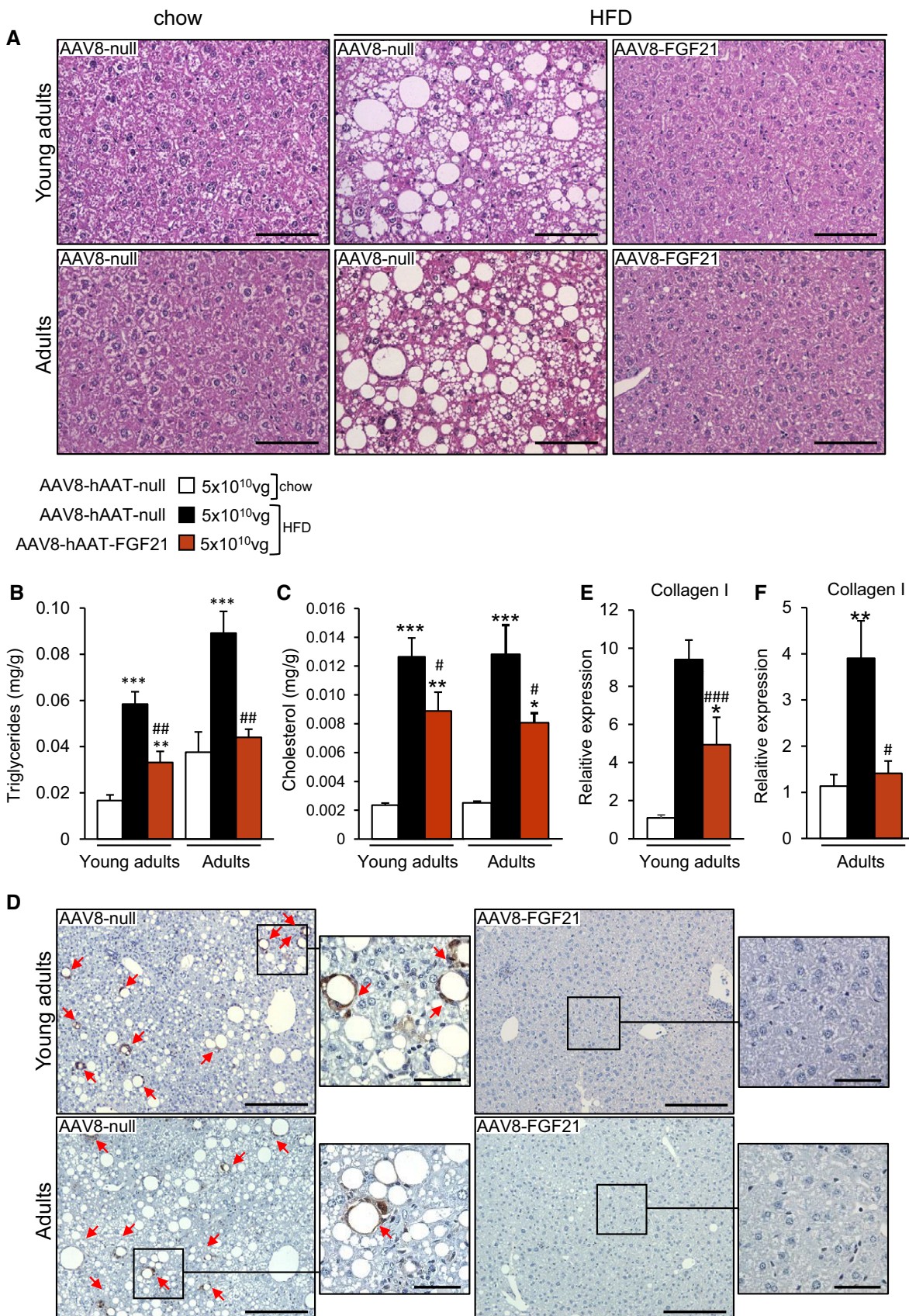

**Figure 4.**

**Figure 4.  Treatment with FGF21-encoding vectors reverses hepatic steatosis and fibrosis.**

A    Representative images of the hematoxylin–eosin staining of liver sections obtained from animals fed a chow or a HFD and administered with either AAV8-hAAT-null or $5 \times 10^{10}$ vg/mouse AAV8-hAAT-FGF21 vectors. HFD clearly induced the deposition of lipid droplets in the liver, and this was reverted by AAV8-hAAT-FGF21 treatment both in young adults and in adults. Scale bars: 100 μm.

B, C  Fed hepatic triglyceride and cholesterol content in the same cohorts of animals.

D    Immunostaining for the macrophage-specific marker Mac2 of liver sections from animals fed a HFD that received either AAV8-hAAT-null or $5 \times 10^{10}$ vg/mouse AAV8-hAAT-FGF21 vectors. Red arrows indicate the presence of crown-like structures. Scale bars: 200 μm and 50 μm (inset).

E, F  Quantification by qRT–PCR of the expression of collagen 1 in the liver in the group of animals that initiated the HFD feeding and received FGF21 vectors as young adults (E) or adults (F).

Data information: All values are expressed as mean ± SEM. In (B, C, E, F), young adults: AAV8-hAAT-null chow ($n = 10$ animals), AAV8-hAAT-null HFD ($n = 8$), AAV8-hAAT-FGF21 HFD $5 \times 10^{10}$ vg ($n = 8$). Adults: AAV8-hAAT-null chow ($n = 7$), AAV8-hAAT-null HFD ($n = 7$), AAV8-hAAT-FGF21 HFD $5 \times 10^{10}$ vg ($n = 7$). In (B, C, E, F), data were analyzed by one-way ANOVA with Tukey's post hoc correction. \*$P < 0.05$, \*\*$P < 0.01$, and \*\*\*$P < 0.001$ versus the chow-fed null-injected group. #$P < 0.05$, ##$P < 0.01$, and ###$P < 0.001$ versus the HFD-fed null-injected group. HFD, high-fat diet.

## Liver overproduction of FGF21 reverses HFD-induced islet hyperplasia and insulin resistance

Null-treated mice fed a HFD for up to 1 year showed normal fasted or fed glycemia (Fig 5A), but were hyperinsulinemic (Fig 5B), suggesting that these mice had developed insulin resistance. In contrast, HFD-fed mice treated with $5 \times 10^{10}$ vg of AAV8-hAAT-FGF21 as young adults or adults were, by the end of the study, normoglycemic and normoinsulinemic in both fasted and fed conditions (Fig 5A and B). In animals treated as adults, administration of the $2 \times 10^{10}$ vg dose also led to normal circulating insulin levels (Fig 5B). The lowest dose, however, did not significantly reduce hyperinsulinemia in any group (Fig 5B). Moreover, HFD-fed animals treated as young adults with AAV8-hAAT-FGF21 vectors showed decreased circulating levels of glucagon compared with HFD-fed null-treated mice (Fig EV2A).

While AAV8-null-treated mice developed islet hyperplasia as a consequence of HFD feeding, the β-cell mass of animals treated with AAV8-hAAT-FGF21 vectors (at the doses of $2 \times 10^{10}$ or $5 \times 10^{10}$ vg/mouse) was similar to that of control mice fed a chow diet (Figs 5C and EV2B). Double immunostaining for insulin and glucagon of pancreatic sections from HFD-fed AAV8-hAAT-FGF21-treated mice showed normal distribution of α- and β-cells in the islets of these animals, with localization of glucagon-expressing cells in the periphery of the islet and of insulin-expressing cells in the core (Fig EV2C).

To confirm the increased insulin sensitivity of FGF21-treated animals, an intraperitoneal insulin tolerance test (ITT) was performed ~2 months post-AAV delivery. As expected, feeding with HFD from the age of 9 or 29 weeks led to a loss of insulin sensitivity in animals injected with AAV8-null (Fig 5D and E). In contrast, animals administered with $5 \times 10^{10}$ vg of AAV8-hAAT-FGF21 as young adults or adults showed greater insulin sensitivity than their age-matched, chow-fed controls (Fig 5D and E). Administration of the intermediate or low dose of AAV8-hAAT-FGF21 also improved insulin sensitivity, irrespective of the age at treatment (Fig 5D and E); the response of these FGF21-treated cohorts was indistinguishable than that of their respective chow-fed control groups (Fig 5D and E).

To evaluate glucose tolerance in FGF21-treated mice, an intraperitoneal glucose tolerance test (GTT) (2 g glucose/kg bw) was performed 10 weeks after AAV administration. HFD-fed animals injected with either null or FGF21-encoding vectors at a dose of $1 \times 10^{10}$ vg/mouse were glucose intolerant and showed markedly increased circulating levels of insulin during the GTT (Fig 5F and G). In contrast, animals treated with $5 \times 10^{10}$ vg/mouse of AAV8-hAAT-FGF21 showed improved glucose clearance when compared to chow-fed control mice (Fig 5F). Insulin levels were indistinguishable between these two experimental groups (Fig 5G). These results further confirmed improved insulin sensitivity in HFD-fed mice treated with $5 \times 10^{10}$ vg/mouse of AAV8-hAAT-FGF21.

## Long-term safety of liver-directed AAV-FGF21 treatment

Pharmacological treatment with FGF21 or transgenic overexpression has been associated with perturbation of bone homeostasis through

**Figure 5.  Treatment with AAV8-hAAT-FGF21 improves insulin sensitivity and glucose tolerance.**

A    Fasted and fed blood glucose levels in mice fed a chow or HFD and injected with either AAV8-hAAT-null or different doses of AAV8-hAAT-FGF21 vectors as young adults or as adults.

B    Fasted and fed serum insulin levels in the same groups of animals as in (A).

C    β-Cell mass in the group of animals that initiated the HFD feeding and received FGF21 vectors as adults.

D, E  Insulin sensitivity was determined in all experimental groups after an intraperitoneal injection of insulin (0.75 units/kg body weight). Results were calculated as the percentage of initial blood glucose levels. During the ITT, basal ($t = 0$ min), peak ($t = 60$ min), and final ($t = 90$ min) glycemia were 166.4 ± 5.7, 37.9 ± 5.2, and 62.4 ± 8.2 mg/dl, respectively, in HFD-fed mice treated as young adults with $5 \times 10^{10}$ vg of AAV8-hAAT-FGF21 vectors.

F    Glucose tolerance was studied in the group of mice that initiated the HFD feeding and received FGF21 vectors as young adults after an intraperitoneal injection of glucose (2 g/kg body weight).

G    Serum insulin levels during the glucose tolerance test shown in (F).

Data information: All data represent the mean ± SEM. In (A, B, D, E), young adults: AAV8-hAAT-null chow ($n = 10$ animals), AAV8-hAAT-null HFD ($n = 8$), AAV8-hAAT-FGF21 HFD $1 \times 10^{10}$ vg ($n = 9$), and $5 \times 10^{10}$ vg ($n = 8$). Adults: AAV8-hAAT-null chow ($n = 7$), AAV8-hAAT-null HFD ($n = 7$), AAV8-hAAT-FGF21 HFD $1 \times 10^{10}$ vg ($n = 7$), $2 \times 10^{10}$ vg ($n = 8$), and $5 \times 10^{10}$ vg ($n = 7$). In (C), adults: AAV8-hAAT-null chow ($n = 5$), AAV8-hAAT-null HFD ($n = 5$), AAV8-hAAT-FGF21 HFD $1 \times 10^{10}$ vg ($n = 4$), $2 \times 10^{10}$ vg ($n = 5$) and $5 \times 10^{10}$ vg ($n = 5$). In (F, G), $n = 6$ animals/group. In (A–G), data were analyzed by one-way ANOVA with Tukey's post hoc correction. \*$P < 0.05$, \*\*$P < 0.01$, and \*\*\*$P < 0.001$ versus the chow-fed null-injected group. #$P < 0.05$, ##$P < 0.01$, and ###$P < 0.001$ versus the HFD-fed null-injected group. HFD, high-fat diet.

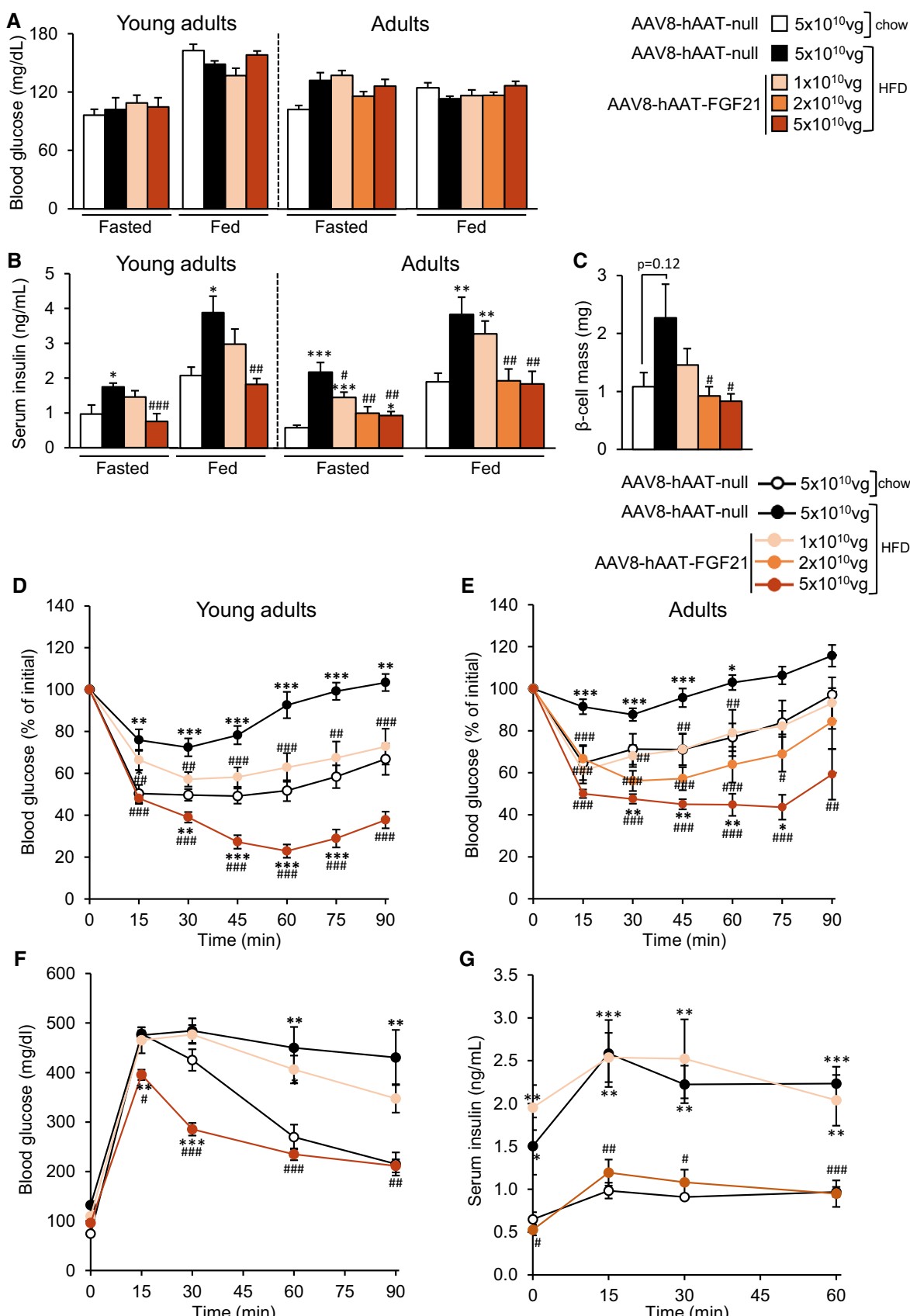

Figure 5.

increased bone resorption, which could cause bone loss (Wei *et al*, 2012; Wang *et al*, 2015; Talukdar *et al*, 2016; Charoenphandhu *et al*, 2017; Kim *et al*, 2017). Given the therapeutic potential of AAV8-hAAT-FGF21 for the treatment of obesity and diabetes, we evaluated the long-term effects of gene transfer on the bones of the animals treated with the highest dose of vector. At the time of sacrifice (~16.5 months of age), the naso-anal length and the tibial length were normal in the animals that were administered with AAV8-hAAT-FGF21 vectors at 9 or 29 weeks of age (Fig EV3A and B). We then examined bone structure by micro-computed tomography (μCT). Analysis of the proximal epiphysis of the tibia revealed no significant differences in the trabecular and cortical bone of mice fed a HFD and administered with $5 \times 10^{10}$ vg AAV8-hAAT-FGF21 in comparison with age-matched mice treated with null vectors. Specifically, no differences were documented in the bone mineral density (BMD) (Fig EV3C), bone mineral content (BMC) (Fig EV3D), bone volume (BV) (Fig EV3E), bone volume/tissue volume ratio (BV/TV) (Fig EV3F), bone surface/bone volume ratio (BS/BV) (Fig EV3G), trabecular number (Tb.N) (Fig EV3H), trabecular thickness (Tb.Th) (Fig EV3I), or trabecular separation (Tb.Sp) (Fig EV3J). Similarly, the analysis of the compact bone at the tibial diaphysis showed no differences in the BMC, BMD, BV, BV/TV, or BS/BV between the HFD-fed null-injected or FGF21-treated groups (Fig EV3K–O).

The pathological effects of FGF21 have been reported to be mediated, at least in part, by increased production of insulin-like growth factor binding protein 1 (IGFPB1) by the liver (Wang *et al*, 2015). In agreement with the lack of bone alterations, high-dose AAV8-hAAT-FGF21 treatment did not lead to an increase in the levels of circulating IGFBP1 protein in animals treated 12 (young adults) or 6 (adults) months earlier when compared to null-injected HFD-fed mice (Fig EV3P). Circulating IGF1 levels were also normal in all experimental groups (Fig EV3Q). Altogether, these results support the safety for bone tissue of AAV-mediated FGF21 gene transfer to the liver.

Long-term feeding (> 60 weeks) with a HFD has been associated with increased incidence of liver neoplasms in C57BL/6J mice (Hill-Baskin *et al*, 2009; Nakagawa, 2015). In our study in animals that initiated the HFD as young adults and maintained it for 60 weeks, we found liver tumors in 66.7% (6/9) of animals injected with null vectors. Animals treated with AAV8-hAAT-FGF21 vectors were protected from HFD-induced development of liver neoplasms: 0% (0/8) of animals treated with the $5 \times 10^{10}$ vg of FGF21-encoding

vectors showed tumors, and the incidence was 40% (4/10) in the cohort treated with the lowest dose ($1 \times 10^{10}$ vg). None (0/11) of the chow-fed mice developed tumors in the same period of time.

### Effects of liver overproduction of FGF21 on ob/ob mice

The ob/ob mice are a well-known model of extreme obesity and insulin resistance (Fellmann *et al*, 2013). To test the anti-obesogenic and anti-diabetic effects of FGF21 gene transfer in this animal model, 8-week-old ob/ob mice were administered IV with either $1 \times 10^{11}$ or $5 \times 10^{11}$ vg/mouse of AAV8-hAAT-FGF21 vectors. Ob/ob animals administered with $5 \times 10^{11}$ vg/mouse of AAV8-null vectors served as controls. While the body weight of null-injected ob/ob mice increased significantly during the follow-up of the study (20 weeks), the body weight of ob/ob animals treated with $5 \times 10^{11}$ vg AAV8-hAAT-FGF21 decreased ~7% during the first 2 weeks after AAV administration and remained stable thereafter (Fig 6A and B and Appendix Fig S5A). Ob/ob mice administered with $1 \times 10^{11}$ vg AAV8-hAAT-FGF21 gained weight during the course of the 20 weeks, but significantly less (10% weight gain) than the AAV8-null-injected cohort (50% weight gain) (Fig 6A and B and Appendix Fig S5A).

Similar to HFD-fed animals, following gene transfer we documented dose-dependent overexpression of FGF21 specifically in the liver (Appendix Fig S5B). This led to secretion of the protein into the bloodstream; similar levels of circulating FGF21 were documented 2 and 5 months following AAV delivery, demonstrating stable production of the protein by the engineered liver (Fig 6C).

In agreement with their lower body weight, ob/ob animals overexpressing FGF21 in the liver showed significantly decreased size of white adipocytes, particularly those animals treated with $5 \times 10^{11}$ vg (Fig 6D). This was parallel with a dose-dependent increase in circulating adiponectin levels (Fig 6E) and decreased WAT inflammation, as evidenced by decreased staining for Mac2 and expression of F4/80 and TNF-α in eWAT (Appendix Fig S5C–E). Noticeably, ob/ob mice treated with $5 \times 10^{11}$ vg showed a remarkable reduction in "crown-like" structures in eWAT (Appendix Fig S5C).

While 7-month-old ob/ob mice showed marked hepatic steatosis, the liver of FGF21-treated ob/ob mice did not show accumulation of lipids in hepatocytes (Fig 6F). This agreed with a 60% reduction in the weight of this organ (Appendix Fig S5F and G) as well as with a

---

**Figure 6.  Reduced obesity and improved insulin sensitivity in ob/ob mice treated with AAV8-hAAT-FGF21 vectors.**

A, B   Follow-up over the course of 5 months of the body weight (A) and body weight gain (B) of ob/ob animals injected at 2 months of age with either $5 \times 10^{11}$ vg/mouse of AAV8-hAAT-null vectors or $1 \times 10^{11}$ or $5 \times 10^{11}$ vg/mouse of AAV8-hAAT-FGF21 vectors.

C   Circulating levels of FGF21 were measured 2 and 5 months after vector administration.

D   Representative images of the hematoxylin–eosin staining of eWAT tissue sections obtained from ob/ob animals injected with either null or FGF21-encoding AAV vectors at $2 \times 10^{10}$ or $5 \times 10^{10}$ vg/mouse. Scale bars: 100 μm.

E   Serum adiponectin levels in all groups.

F   Representative images of the hematoxylin–eosin staining of liver tissue sections obtained from ob/ob animals injected with either null or FGF21-encoding AAV vectors at $2 \times 10^{10}$ or $5 \times 10^{10}$ vg/mouse. Scale bars: 200 μm.

G   Fed blood glucose levels.

H   Fed serum Insulin levels.

I   Insulin tolerance test after intraperitoneal injection of insulin at a dose of 0.75 units/kg body weight. Results were calculated as the percentage of initial blood glucose. Ob/ob mice treated with AAV8-hAAT-FGF21 vectors at any of the doses showed greater insulin sensitivity.

Data information: All data represent the mean ± SEM. In (A–C, G–I), AAV8-hAAT-null (*n* = 10 animals), AAV8-hAAT-FGF21 $1 \times 10^{11}$ vg (*n* = 10), and $5 \times 10^{11}$ vg (*n* = 9). In (A–C, E, G–I), data were analyzed by one-way ANOVA with Tukey's post hoc correction. *$P < 0.05$, **$P < 0.01$, and ***$P < 0.001$ versus null-injected ob/ob group.

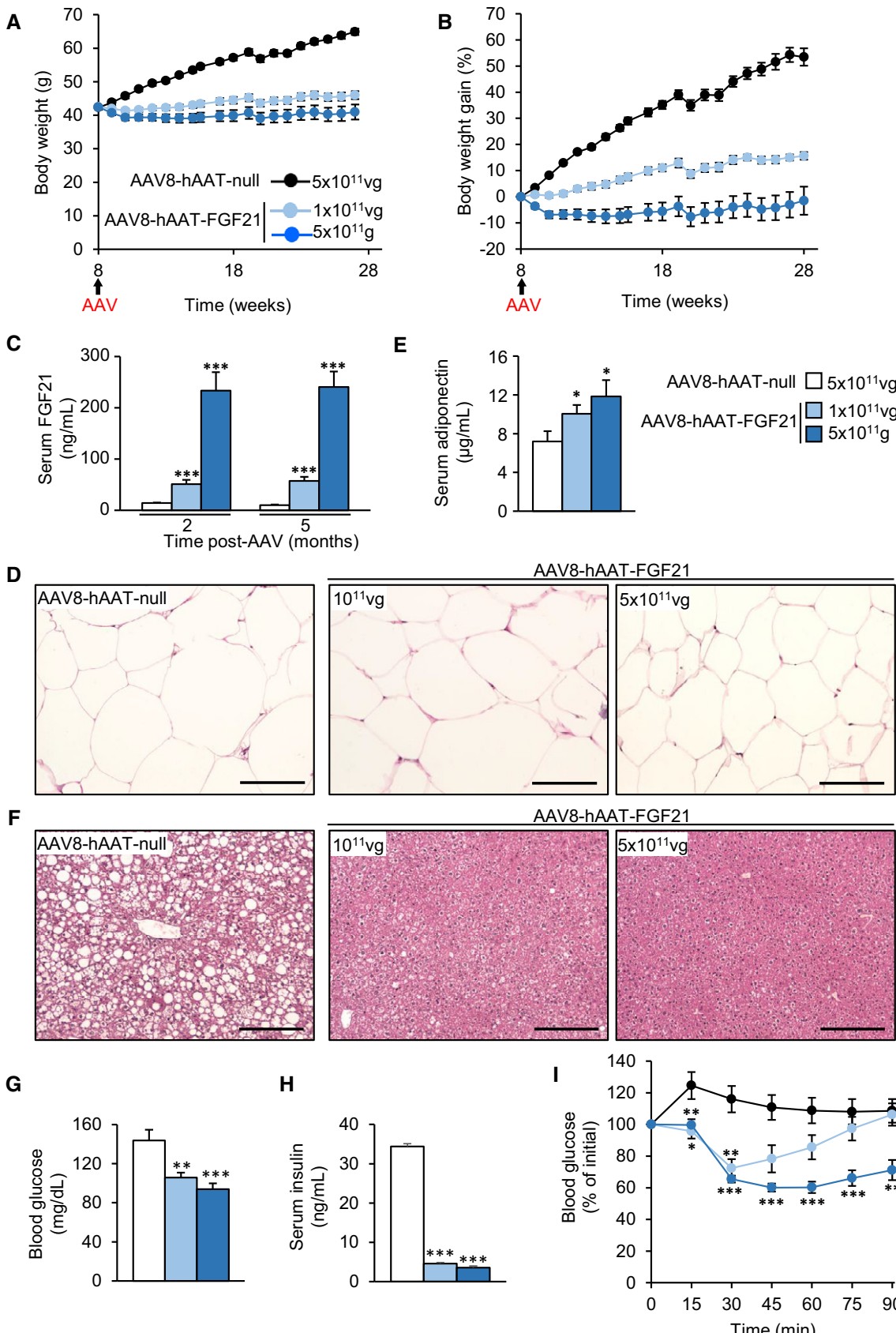

**Figure 6.**

marked reduction in the total liver triglyceride and cholesterol content (Appendix Fig S5H and I) in ob/ob mice receiving therapeutic vectors. Ob/ob animals treated with both doses of AAV8-hAAT-FGF21 also showed decreased fed glycemia, and their insulinemia in the fed state was reduced by ~70% (Fig 6G and H). Treatment with AAV8-hAAT-FGF21 improved insulin sensitivity in comparison with AAV8-hAAT-null-treated ob/ob mice (Fig 6I), confirming the observations made in HFD-fed animals.

We evaluated whether the decrease in circulating glucose levels observed in ob/ob mice after AAV-FGF21 treatment resulted from suppression of hepatic gluconeogenesis by measuring the expression by qPCR of phosphoenolpyruvate carboxykinase (PEPCK) and glucose-6-phosphatase (G6Pase). No changes in the expression of these enzymes were observed in the liver of AAV-FGF21-treated ob/ob mice, except for the animals treated with $1 \times 10^{11}$ vg of AAV8-hAAT-FGF21 that showed increased PEPCK expression (Appendix Fig S6A and B). These results suggested that AAV-mediated long-term expression of FGF21 in the liver, and the subsequent increase in circulating FGF21, did not lower glucose by inhibiting hepatic glucose production.

The glucose-lowering effects of FGF21 have also been attributed to increased glucose uptake by adipocytes and enhanced energy expenditure (Kharitonenkov *et al*, 2005; Xu *et al*, 2009b; Hondares *et al*, 2010; Ding *et al*, 2012; Camacho *et al*, 2013; Emanuelli *et al*, 2014; Samms *et al*, 2015). Thus, we assessed in different pads of adipose tissue (iWAT, eWAT, and iBAT) the expression of key components of the glucose uptake machinery by qPCR, such as the glucose transporters *Glut1* and *Glut4*, the glucose phosphorylating enzymes hexokinase I and II (HKI and HKI), and UCP1 in the case of iBAT. In AAV8-FGF21-treated ob/ob mice, the expression of *Glut1* was increased in iWAT and iBAT (Appendix Fig S6C) and that of *Glut4* was increased in eWAT, iWAT, and iBAT (Appendix Fig S6D). HKI and HKII were upregulated only in iBAT (Appendix Fig S6E and F). Moreover, UCP1 expression was increased in the iBAT of ob/ob mice treated with the high dose of AAV8-FGF21 vectors (Appendix Fig S6G). Altogether, these results suggest that the long-term amelioration of glycemia observed in ob/ob mice following treatment with AAV-FGF21 vectors probably results from increased glucose uptake by white and brown adipocytes and enhanced thermogenesis in iBAT.

## Reversion of obesity and insulin resistance by AAV8-mediated gene transfer of FGF21 to the eWAT of ob/ob mice

To explore whether it was possible to achieve the same degree of therapeutic benefit through production of FGF21 by another tissue besides the liver, we engineered the epididymal white adipose tissue pad by intra-depot administration of FGF21-encoding AAV8 vectors. Eleven-week-old ob/ob mice received an intra-eWAT injection of $1 \times 10^{10}$, $5 \times 10^{10}$, $2 \times 10^{11}$, or $1 \times 10^{12}$ vg/mouse of AAV8 in which murine optimized FGF21 was under the transcriptional control of the ubiquitous CAG promoter. To avoid expression of the transgene in other main organs for which AAV8 shows strong tropism, such as liver and heart (Gao *et al*, 2002; Zincarelli *et al*, 2008; Wang *et al*, 2010), we took advantage of microRNAs (miRs). Target sequences for miR-122a and miR-1, which selectively detarget transgene expression from liver and heart when included into AAV vectors (Jimenez *et al*, 2013; Mallol *et al*, 2017), were added in tandem repeats of four copies to the 3′-UTR of the FGF21 expression cassette (AAV8-CAG-FGF21-dmiRT). Animals injected with vectors lacking any coding sequence (AAV8-CAG-null) at a dose of $1 \times 10^{12}$ vg/mouse were used as controls.

While ob/ob mice injected with null vectors continued to gain weight during the 14-week follow-up period (~45% weight gain), there was a clear dose-dependent moderation of the weight gain in cohorts treated with FGF21-encoding vectors (~25, ~18, ~8% for doses $1 \times 10^{10}$, $5 \times 10^{10}$, $2 \times 10^{11}$ vg/mouse, respectively, Fig 7A and B). Moreover, animals injected with the highest dose of AAV8-CAG-FGF21-dmiRT showed an initial loss of weight of approximately ~13% and by the end of the experiment had returned to their initial body weight (Fig 7B).

Gene transfer to the eWAT resulted in a dose-dependent increase in the levels of FGF21 in serum (Fig 7C). Circulating FGF21 rose mainly due to expression of the transgene in the eWAT; limited expression was observed in iWAT and only at the highest doses of vector used (Fig 7D). In agreement with our previous work (Jimenez *et al*, 2013; Mallol *et al*, 2017), the inclusion of the microRNA target sequences efficiently prevented expression in heart and liver; only marginal expression was observed in the liver at the highest dose used (data not shown).

Similar to the observations made in ob/ob mice in which FGF21 gene transfer was targeted to the liver, ob/ob mice that received

---

**Figure 7. Intra-eWAT administration of FGF21 vectors in ob/ob mice.**

A, B Progression of body weight (A) and body weight gain (B) in ob/ob animals treated at 11 weeks of age with an intra-WAT injection of either AAV8-CAG-null vectors ($1 \times 10^{12}$ vg/mouse) or AAV8-CAG-FGF21-dmiRT vectors at four different doses ($1 \times 10^{10}$, $5 \times 10^{10}$, $2 \times 10^{11}$, $1 \times 10^{12}$ vg/mouse).

C Serum levels of FGF21 measured at the end of the 14-week follow-up period.

D Quantitative PCR analysis of FGF21 expression in the eWAT and iWAT fat pads of the same cohorts as in (A). The qPCR was performed with primers that specifically detected vector-derived FGF21 mRNA.

E, F Representative images of the hematoxylin–eosin staining of (E) eWAT and (F) liver tissue sections obtained from ob/ob animals injected intra-eWAT either null or FGF21-encoding AAV8 vectors at all doses tested. Scale bars: 100 μm for eWAT (E) and 200 μm for liver (F).

G Glycemia in the fed state.

H Insulinemia in the fed state.

I Insulin tolerance test after intraperitoneal injection of insulin at a dose of 0.75 units/kg body weight. Results were calculated as the percentage of initial blood glucose.

Data information: All values are expressed as mean ± SEM. In (A, B, I) AAV8-hAAT-null ($n = 7$ animals), AAV8-hAAT-FGF21 $1 \times 10^{10}$ vg ($n = 6$), $5 \times 10^{10}$ vg ($n = 6$), $2 \times 10^{11}$ vg ($n = 7$), and $1 \times 10^{12}$ vg ($n = 8$). In (C, D, G, H), AAV8-hAAT-null ($n = 7$), AAV8-hAAT-FGF21 $1 \times 10^{10}$ vg ($n = 6$), $5 \times 10^{10}$ vg ($n = 4$), $2 \times 10^{11}$ vg ($n = 7$), and $1 \times 10^{12}$ vg ($n = 8$). In (A–D, G–I), data were analyzed by one-way ANOVA with Tukey's post hoc correction. *$P < 0.05$, **$P < 0.01$ and ***$P < 0.001$ versus the null-injected group.

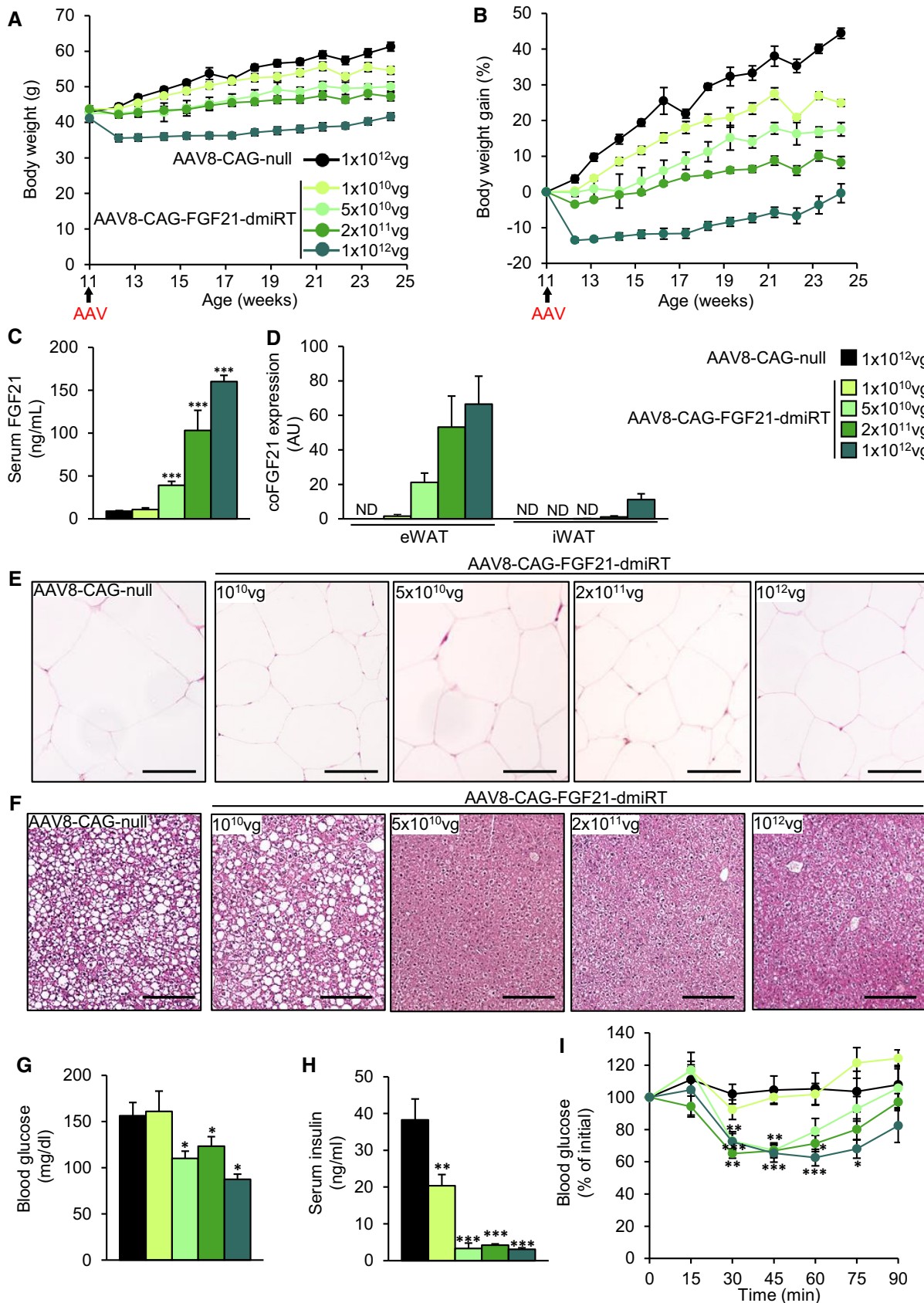

Figure 7.

intra-eWAT injections of AAV8-CAG-FGF21-dmiRT vectors showed a reduction in the size of white adipocytes of the epididymal pad (Fig 7E). Circulating adiponectin levels also increased with dose (Appendix Fig S7A). eWAT inflammation, evaluated through Mac2 staining, was also reduced as a function of the dose of vector, as did the expression of the macrophage marker F4/80 (Appendix Fig S7B and C).

The liver of ob/ob mice injected with null vectors or the lowest dose of AAV8-CAG-FGF21-dmiRT showed accumulation of lipid droplets in hepatocytes (Fig 7F). The administration of doses of $5 \times 10^{10}$ vg/mouse or higher of FGF21-encoding vectors completely prevented the development of hepatic steatosis (Fig 7F), which correlated with the weight of the organ (Appendix Fig S7D) and its total triglyceride and cholesterol content (Appendix Fig S7E and F). Further evidence that the dose of $5 \times 10^{10}$ vg/mouse represented a threshold for therapeutic efficacy came from the analysis of glycemia and insulinemia. While the dose of $1 \times 10^{10}$ vg/mouse did not modify the levels of blood glucose in the fed state and only partially reduced insulin levels, doses of $5 \times 10^{10}$ vg/mouse and higher completely normalized glycemia and insulinemia (Fig 7G and H). Accordingly, the three highest doses of AAV8-CAG-FGF21-dmiRT injected in the eWAT greatly improved insulin sensitivity in comparison with AAV8-null-injected ob/ob mice, while the lowest dose had no effect on the ITT (Fig 7I). Altogether, this study confirmed the therapeutic potential of overexpressing FGF21 in adipose tissue.

### The skeletal muscle as a source of circulating FGF21

Skeletal muscle (Skm) is a readily accessible tissue and has been used to produce secretable therapeutic proteins (Haurigot *et al*, 2010; Callejas *et al*, 2013; Jaén *et al*, 2017). To explore whether the Skm could represent a viable source of circulating FGF21, AAV vectors of serotype 1, which show a high tropism for Skm (Chao *et al*, 2000; Wu *et al*, 2006; Lisowski *et al*, 2015), carrying murine optimized FGF21 under the control of the CMV promoter were used (AAV1-CMV-FGF21). Vectors were injected at a dose of $5 \times 10^{10}$ vg/muscle to the quadriceps, gastrocnemius, and tibialis cranialis of both legs (total dose, $3 \times 10^{11}$ vg/mouse) of 8-week-old C57Bl6 mice. Control animals were injected with AAV1-null vectors at the same dose. The use of healthy mice fed a standard diet further allowed us to evaluate the long-term safety of FGF21 gene therapy.

Eleven-month-old animals injected with FGF21-encoding vectors at 8 weeks of age showed a marked increase in circulating FGF21 (Fig 8A), which was parallel to high levels of expression of

vector-derived FGF21 in the three injected muscles (Fig 8B). In agreement with previous reports, this combination of vector serotype, promoter, and route of administration did not lead to expression of the transgene in the liver (Fig 8B).

At the end of the ~10-month follow-up period, mice injected intramuscularly with AAV1-CMV-FGF21 maintained the body weight they had at the initiation of the study and were ~38% slimmer than controls, which steadily increased their weight as animals aged (Fig 8C). While the weight of the muscles was barely affected by FGF21 gene transfer, the weight of the white and brown depots as well as the liver was considerably reduced (Fig 8D). Indeed, the weight of the WAT pads analyzed was reduced by > 50% (Fig 8D). Moreover, mice treated with AAV1-CMV-FGF21 showed a marked reduction in the hepatic total triglyceride content (Fig 8E). No changes in hepatic cholesterol levels were observed (Fig 8F). As opposed to null-injected animals, animals treated with AAV1-CMV-FGF21 showed normoglycemia (data not shown) and reduced insulinemia when they were approximately 1 year old (Fig 8G). Accordingly, FGF21-treated mice showed markedly improved insulin sensitivity at the end of the study (Fig 8H). Altogether, this study demonstrates that administration of AAV vectors that leads to therapeutically relevant levels of circulating FGF21 is safe in the long-term in healthy and may be used to counteract the increase in body weight and insulin resistance associated with aging.

## Discussion

The present work provides the first evidence of long-term counteraction of obesity and insulin resistance upon a one-time administration of a gene therapy AAV vector encoding FGF21. First, we took advantage of AAV8 vectors and a liver-specific promoter to overexpress a codon-optimized FGF21 coding sequence in the liver, the main tissue from where circulating endogenous FGF21 is derived (Markan *et al*, 2014). This resulted in a sustained increase in FGF21 levels in the bloodstream. We demonstrated disease reversal for > 1 year by the AAV8-hAAT-FGF21 approach in HFD-fed mice—the model that most closely resembles the metabolic characteristics of human obesity and insulin resistance—treated either as young adults or when older than 9 months. To the best of our knowledge, this is the longest follow-up ever reported for an FGF21-based treatment in HFD-fed mice and demonstrates both the efficacy and the safety of our therapeutic approach. Remarkably, the efficacy of FGF21 gene transfer to the liver was confirmed in the genetically obese ob/ob mouse model. Moreover, gene transfer directly to

▶

**Figure 8.  Gene transfer of FGF21 to the skeletal muscle of healthy animals.**

A    Circulating levels of FGF21 measured 40 weeks after injection of $3 \times 10^{11}$ vg/mouse of either AAV1-CMV-null or AAV1-CMV-FGF21 vectors to the skeletal muscle of healthy animals fed a chow diet.

B    AAV-derived FGF21 expression in the muscles and liver of healthy animals injected intramuscularly with AAV1-CMV-null or AAV1-CMV-FGF21 vectors.

C    Evolution of the body weight in the 40-week follow-up period.

D    Wet tissue weight of different muscles, adipose pads, and liver.

E, F    Hepatic triglyceride and cholesterol content in the fed state.

G    Fed serum insulin levels.

H    Insulin sensitivity assessed through intraperitoneal injection of insulin (0.75 units/kg body weight) and represented as percentage of initial blood glucose.

Data information: All values are expressed as mean ± SEM. In (A–H), AAV1-CMV-null (*n* = 5 animals) and AAV1-CMV-FGF21 (*n* = 7). In (A–H), data were analyzed by unpaired Student's *t*-test. *P < 0.05, **P < 0.01, and ***P < 0.001 versus the null-injected group.

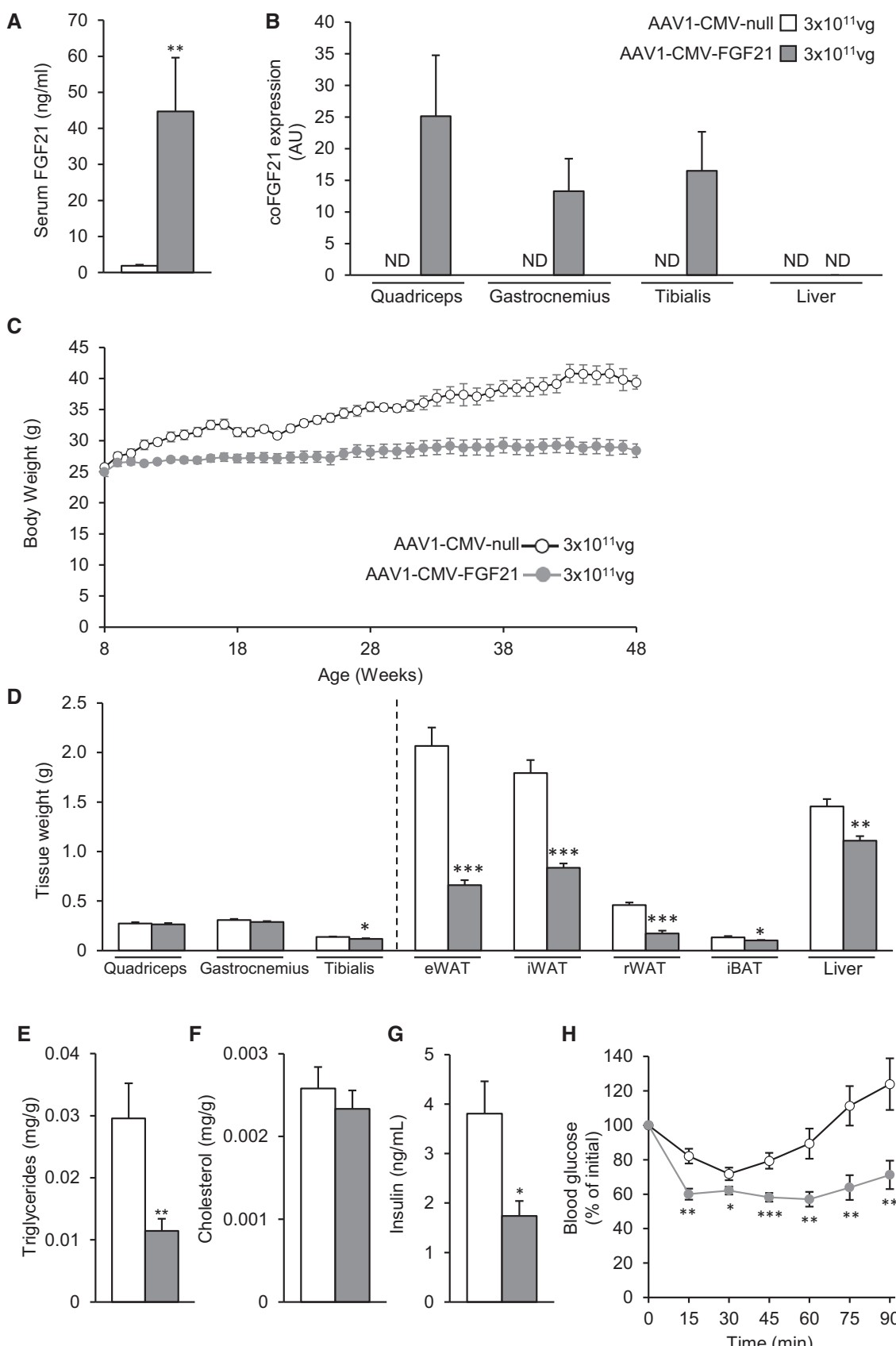

Figure 8.

epididymal white adipose tissue of ob/ob mice afforded the same degree of therapeutic benefit, and administration of FGF21-encoding AAV vectors to the skeletal muscle counteracted age-associated body weight gain and insulin resistance.

Intravascular administration of FGF21-coding AAV8 vectors targeting the liver several weeks after the initiation of feeding with a HFD led to a dose–response reduction in body weight. Importantly, once that body weight of HFD-fed obese animals was normalized, it remained stable for the whole follow-up period and, in the case of the animals that received the high dose of vector, it never reached values below the weight at the initiation of the study. The loss of body weight was parallel to a decrease in the weight of the adipose tissue pads and in adipocyte size. In agreement with the reduction in the size of the adipose depots, the levels of the adipokine adiponectin and leptin were normalized, and no WAT inflammation was detected. Similar observations were described in transgenic mice overexpressing FGF21 (Tg-FGF21) (Kharitonenkov *et al*, 2005; Inagaki *et al*, 2007) and in animals treated pharmacologically with native FGF21 or FGF21 analogues/mimetics (Coskun *et al*, 2008; Berglund *et al*, 2009; Xu *et al*, 2009a; Adams *et al*, 2012a,b; Foltz *et al*, 2012; Hecht *et al*, 2012; Smith *et al*, 2013; Emanuelli *et al*, 2014; Zhang & Li, 2015; Markan & Potthoff, 2016; So & Leung, 2016; Talukdar *et al*, 2016; BonDurant *et al*, 2017; Stanislaus *et al*, 2017). Importantly, treatment with FGF21 vectors also prevented body weight gain and adipose tissue hypertrophy and inflammation in ob/ob mice.

The clear effect of FGF21 on body weight in HFD-fed treated animals was observed despite an increase in food intake, in agreement with previous reports (Kharitonenkov *et al*, 2005; Coskun *et al*, 2008; Inagaki *et al*, 2008; Laeger *et al*, 2017). Since FGF21 has been shown to be able to cross the blood–brain barrier (Hsuchou *et al*, 2007), and circulating FGF21 levels were increased following gene transfer, the effect of the treatment on food intake could be the consequence of a direct effect of FGF21 on the central control of food intake. Alternatively, FGF21 may have indirectly affected food intake by increasing energy expenditure in treated animals *(vide infra)*, who compensate for this increase by eating more.

Possible explanations for the reduction in body weight may be provided by the increase in energy expenditure and locomotor activity documented in AAV8-hAAT-FGF21-treated animals, also observed previously after pharmacological FGF21 treatment (Coskun *et al*, 2008; Xu *et al*, 2009a). However, previous studies have described a decrease in physical activity and in basal core temperature in both fasted transgenic mice overexpressing murine FGF21 specifically in the liver under control of the *ApoE* promoter (ApoE-mFGF21) and in lean mice fed a ketogenic diet (KD) (Inagaki *et al*, 2007; Bookout *et al*, 2013). This led to the hypothesis that FGF21 induced torpor. Other studies have reported observations in agreement with our results and opposed to those reported previously (Inagaki *et al*, 2007; Bookout *et al*, 2013). In a different line of FGF21 transgenic mice, in which overexpression of human FGF21 was also driven by the *ApoE* promoter (ApoE-hFGF21), no differences in body temperature were observed when mice were fed a HFD (Kharitonenkov *et al*, 2005), which suggested that the effects of FGF21 may be dependent on nutrient context (Solon-Biet *et al*, 2016). In agreement with this hypothesis, treatment of HFD-fed WT mice with recombinant FGF21 also augmented body temperature and physical activity (Coskun *et al*, 2008; Xu *et al*, 2009a). We also

observed an increase in locomotor activity following intravascular administration of AAV8-hAAT-FGF21 vectors to HFD-fed mice which, together with the observations of previous studies argue against the possibility of FGF21 being a torpor-promoting factor.

The raise in energy expenditure following FGF21 gene transfer could be the consequence of increased non-shivering thermogenesis in BAT, as suggested by the reduction in lipid deposition and the increment in UCP1 protein levels in BAT. This agrees with previously published data in small animals treated with recombinant native FGF21 or with FGF21 analogues/mimetics which demonstrated increased UCP1 in BAT and enhanced energy expenditure as a consequence of the treatment (Owen *et al*, 2014; Douris *et al*, 2015; Samms *et al*, 2015; Véniant *et al*, 2015). However, there is currently controversy on whether FGF21 effects are UCP1 dependent or not. It has been described that treatment of UCP1 knockout mice with a long-acting FGF21 analogue increased energy expenditure despite the lack of UCP1 (Véniant *et al*, 2015). In contrast, in another study in UCP1-deficient mice FGF21 failed to increase metabolic rate (Samms *et al*, 2015). Supporting a UCP1-independent mechanism, the effects of FGF21 administration to mice have been reported to be maintained after ablation of iBAT (Camporez *et al*, 2013; Emanuelli *et al*, 2014; Bernardo *et al*, 2015). In these studies, only the iBAT pad was surgically excised, what may in turn result in increased compensatory thermogenic activity in the remaining BAT depots of the animal and/or in increased appearance of thermogenic-competent beige adipocytes in subcutaneous WAT. In agreement with this possibility, an increase in UCP1 expression and protein content were detected in the iWAT of mice treated with FGF21 upon surgical removal of iBAT, compared with the levels observed in sham-operated FGF21-treated animals (Bernardo *et al*, 2015). Therefore, we believe it cannot be ruled out that that FGF21 effects on energy expenditure are UCP1-dependent.

An FGF21-mediated induction of UCP1 has also been described for iWAT (Hondares *et al*, 2010; Adams *et al*, 2012a,b; Fisher *et al*, 2012; Emanuelli *et al*, 2014; Douris *et al*, 2015). Nevertheless, in AAV8-hAAT-FGF21-treated animals, we did not observe an induction of browning in iWAT. In agreement with our results, treatment of HFD-fed mice or obese cynomolgus primates with FGF21 analogues also failed to induce browning of iWAT, but mediated considerable loss of body weight (Véniant *et al*, 2015; Talukdar *et al*, 2016). In addition, our results also support the idea that FGF21 effects on glucose homeostasis *(vide infra)* are independent from UCP1 induction in WAT (Samms *et al*, 2015; Véniant *et al*, 2015). Altogether, these results indicate that browning of iWAT is not necessary for FGF21 anti-obesogenic and anti-diabetic effects.

Two recent studies demonstrated alternative mechanisms by which the subcutaneous adipose tissue could improve energy homeostasis such as the creatine-driven cycle (Kazak *et al*, 2015) and the Serca2b-dependent calcium cycling (Ikeda *et al*, 2017). Treatment with AAV8-hAAT-FGF21 vectors increased the expression of Phospho1 in iWAT suggesting that UCP1-independent mechanisms may also contribute to the enhancement of energy expenditure mediated by treatment with AAV-FGF21 vectors.

Non-alcoholic fatty liver disease and NASH are strongly associated with insulin resistance and T2D in humans (Kitade *et al*, 2017) and constitute main targets in the development of anti-diabetic treatments. In our study, feeding of mice with a HFD for over 10 months

resulted in increased weight of the liver and marked alteration of its structure, with presence of large lipid vacuoles and fibrosis in the liver parenchyma as well as marked inflammation. Liver FGF21 gene therapy led to complete correction of liver alterations, and the liver of AAV8-hAAT-FGF21-treated HFD-fed animals resembled that of mice fed a chow diet. Similar observations were made in ob/ob mice treated with intravascular AAV8-hAAT-FGF21. Altogether, these results underscore the potential of FGF21 gene transfer to the liver to counteract this pro-diabetogenic condition.

As a result of the improvements observed in white and brown adipose tissues and in liver, both HFD-fed and ob/ob mice treated with liver-targeted FGF21 gene therapy showed a remarkable increase in insulin sensitivity, clearly demonstrated by the correction of hyperinsulinemia and the improved performance in the ITT and GTT. Noticeably, HFD-fed mice treated with the lowest dose of vector, which did not prevent weight gain, also showed an improvement in insulin sensitivity, suggesting that not all the positive effects of FGF21 on glucose metabolism are dependent on weight loss. Similar effects on glucose homeostasis have previously been reported in insulin-resistant rodents and in obese diabetic non-human primates (Kharitonenkov *et al*, 2007; Berglund *et al*, 2009; Hale *et al*, 2012; Hecht *et al*, 2012; Véniant *et al*, 2012; Adams *et al*, 2013; Camporez *et al*, 2013; Emanuelli *et al*, 2014; Charoenphandhu *et al*, 2017; Stanislaus *et al*, 2017). Clinical studies with FGF21 analogues conducted so far, however, have failed to show improvements on glycaemia (Gaich *et al*, 2013; Talukdar *et al*, 2016; Kim *et al*, 2017), although studies were likely too short to fully reproduce observations made in animal models. It remains to be determined whether gene therapy, which leads to constant production of the protein and steady circulating levels, can improve glycaemia and insulin sensitivity in humans.

As an alternative to the liver, we explored the possibility of using white adipose tissue as a secretory organ for FGF21. The high secretory capacity of this organ may be exploited for the development of new gene therapy strategies for diseases in which supply of the therapeutic agent into the bloodstream is needed for treatment (Jimenez *et al*, 2013). Results confirmed that AAV-engineered WAT could act as a pump to secrete FGF21 to the circulation and mediate the same therapeutic efficacy achieved when targeting the liver. We observed that following intra-eWAT administration of the highest dose, some of the vector leaked to the bloodstream and transduced iWAT, although at much lower levels. Actually, the minimal efficacious dose ($5 \times 10^{10}$ vg/mouse) did not lead to iWAT transduction, confirming that expression of FGF21 in this organ is not a requirement for therapeutic effect. Expression in other potentially important target organs for AAV8 transduction, such as liver and heart, was efficiently prevented by inclusion of target sequences for microRNAs strongly expressed in these organs (Shingara *et al*, 2005; Qiao *et al*, 2011). Therefore, WAT is an attractive alternative target site, particularly for patients not eligible for liver-directed gene transfer because of underlying hepatic diseases, such as cirrhosis or liver cancer. WAT is also an easily accessible organ and there is the possibility of easy surgical removal in case of adverse events.

Additionally, we also tested the ability of the skeletal muscle to produce and secrete FGF21. This study not only confirmed the skeletal muscle as another possible target organ for FGF21 gene therapy but also demonstrated a very exciting finding of our work:

that gene transfer of FGF21 to healthy animals fed a chow diet was safe and could prevent the weight gain and insulin resistance developed as animals' age. Although further studies are warranted, this observation opens the spectrum of potential indications for FGF21 gene therapy, especially for healthy aging.

The potential of FGF21 to treat obesity and insulin resistance has prompted the pharmaceutical industry to develop a myriad of analogues and mimetics that have the same biological actions than endogenous FGF21 but improved pharmacokinetic properties (Zhang & Li, 2015; So & Leung, 2016). Gene therapy can offer some advantage over these designed molecules. On the one hand, there is the possibility of achieving therapeutic levels of a protein following a single administration of the gene therapy product, with obvious implications for patient management. Despite the fact that long-acting FGF21 molecules have been developed, they still require periodic administrations that are inconvenient for patients and may compromise in some cases therapeutic adherence. But most importantly, gene therapy allows one to work with the wild-type protein, that is recognized as own by the immune system and signals through the canonical FGF21 signaling pathways, avoiding unspecific biological responses. Regardless of the route of administration, long-term steady levels of circulating FGF21 were obtained in mice treated with AAV-FGF21 vectors, supporting the absence of anti-transgene immune responses in these animals. In contrast, humoral responses have been observed shortly after administration of FGF21 analogues to non-human primates and humans (Adams *et al*, 2013; Gaich *et al*, 2013; Kim *et al*, 2017; Stanislaus *et al*, 2017), and the implications of these responses to long-term drug efficacy are yet unknown. In addition, those FGF21-class molecules that act as receptor agonists, both in a Klotho-dependent or in Klotho-independent manner, may have unpredictable biological effects *in vivo*.

The two main safety concerns to be considered for a chronic treatment with FGF21 are possible deleterious effects on bone homeostasis and, given the growth factor nature of FGF21, the potential for tumorigenesis (Kharitonenkov *et al*, 2005; Kharitonenkov & DiMarchi, 2017). Tg-FGF21 mice expressing the transgene specifically in the liver are shorter, and bone loss has been reported in Tg-FGF21 mice as well as in animals and humans treated with recombinant native FGF21 protein or FGF21 analogues/mimetics (Wei *et al*, 2012; Wang *et al*, 2015; Talukdar *et al*, 2016; Charoenphandhu *et al*, 2017; Kim *et al*, 2017). Despite the fact that the levels of circulating FGF21 achieved in our study in HFD-fed mice treated with the highest dose were in the same range as those observed in Tg-FGF21 mice (Inagaki *et al*, 2008), we did not observe any differences in the naso-anal or tibial length in AAV8-hAAT-FGF21-treated animals, irrespective of the age at treatment. It is worth noticing that the group of animals that were treated as young adults received the vectors at an age at which closure of the bone growth plate has not yet occurred (Kilborn *et al*, 2002). Our mice were of the same strain than the Tg-FG21 reported to be shorter, then the striking differences on bone growth may be due to effects caused during embryonic development and/or early post-natal life of transgenic animals. Alternatively, it has been postulated that the bone phenotype observed in TgFGF21 might not be a direct consequence of FGF21 action but rather of the chronic, FGF21-induced negative energy balance present in these animals (Kharitonenkov & DiMarchi, 2017). On the other hand, there were no signs of

trabecular and cortical bone loss following treatment of HFD-fed mice with FGF21-encoding AAV8 vectors. There is currently controversy as whether FGF21 has deleterious effects on bone mineral density. Tg-FGF21 mice have reduced bone mass (Wei *et al*, 2012; Wang *et al*, 2015). Short-term pharmacological treatment with native FGF21 or analogues/mimetics caused bone loss and/or alteration of markers of bone turnover in rodents and humans (Wei *et al*, 2012; Wang *et al*, 2015; Talukdar *et al*, 2016; Charoenphandhu *et al*, 2017; Kim *et al*, 2017). In contrast, a very recent study described no effect of short-term FGF21 treatment on bone mass in HFD-fed mice, suggesting that FGF21 is not critical for bone homeostasis in rodents (Li *et al*, 2017).

Regarding tumor formation, although the number of animals used in our studies was low, the striking difference in the incidence of liver neoplasms between animals injected with null vectors or therapeutic vectors actually argues in favor of a protective role of FGF21 against malignancies. FGF21 gene transfer to the liver did not only not cause tumors *per se* but also prevented the formation of tumors induced by long-term HFD feeding. In agreement with these findings, no signs of pathological proliferation were detected in the liver of two different transgenic mouse models overexpressing FGF21 from birth (Kharitonenkov *et al*, 2005; Huang *et al*, 2006) or in mice treated with recombinant native FGF21 (Adams *et al*, 2012a,b). Moreover, overexpression of FGF21 specifically in the liver in transgenic mice or treatment with native FGF21 delayed the appearance of chemically induced tumors (Huang *et al*, 2006; Xu *et al*, 2015).

Noticeably, the levels of FGF21 in circulation necessary to mediate beneficial therapeutic effects were very similar independently of the target organ for gene transfer and were around 25–50 ng/ml, which is much lower than the peak concentrations achieved in clinical trials after periodic administration of FGF21 analogues (Dong *et al*, 2015; Talukdar *et al*, 2016). This indicates that constant low levels of serum FGF21 could be more efficacious. Importantly, these levels of circulating FGF21 did not produce bone toxicity in mice and prevented the development of liver tumors associated with long-term HFD feeding.

With its excellent record of efficacy and safety, and the approval in Europe of the first *in vivo* gene therapy product in 2012 (Büning, 2013)—with others under way—AAV-mediated gene therapy has broadened its range of applications from monogenic to non-hereditary diseases, such as diabetes. There is considerable preclinical and clinical experience with liver and muscle-directed gene transfer for a variety of hereditary conditions, which provide support to the feasibility of approaches such as those proposed here for the treatment of complex metabolic diseases in humans. In this work, we have developed gene therapy approaches for obesity and insulin resistance based on the use of AAV vectors encoding FGF21. One-time administration of these vectors to obese animals enabled a long-lasting increase in FGF21 levels in circulation, which resulted in sustained counteraction of obesity, adipose tissue inflammation, insulin resistance, and NASH in the absence of adverse events. In healthy animals, this approach was safe and promoted healthy aging. Our results constitute the basis to support the future clinical translation of FGF21 gene transfer to treat T2D, obesity, and related comorbidities. Nevertheless, studies that investigate the long-term safety and efficacy of the approach in large animals, including non-human primates, are mandatory before moving AAV-FGF21-mediated gene

therapy to the clinic for the treatment of these highly prevalent diseases.

# Materials and Methods

### Animals

Eight-, nine-, or 29-week-old male C57BL/6J mice and 8- or 11-week-old B6.V-*Lep^ob*/OlaHsd (ob/ob) mice were used. Mice were kept in a specific pathogen-free facility (SER-CBATEG, UAB) and maintained under a light–dark cycle of 12 h at 22°C. Mice were fed *ad libitum* with a standard diet (2018S Teklad Global Diets®, Envigo) or a high-fat diet (TD.88137 Harlan Teklad). When stated, mice were fasted for 16 h. For tissue sampling, mice were anesthetized with inhalational anesthetic isoflurane (IsoFlo®, Abbott Laboratories, Abbott Park, IL, USA) and decapitated. Tissues of interest were excised and kept at −80°C or in formalin until analysis. Animal care and experimental procedures were approved by the Ethics Committee in Animal and Human Experimentation of the Universitat Autònoma de Barcelona.

### Recombinant AAV vectors

An AAV expression cassette was obtained by cloning, between the ITRs of AAV2, the murine codon-optimized FGF21 coding sequence (coFGF21) under the control of either: (i) the human α1-antitrypsin promoter (hAAT); (ii) the cytomegalovirus (CMV) promoter, or (iii) the early CMV enhancer/chicken beta actin (CAG) promoter with the addition of four tandem repeats of miRT122a (5′CAAACACCATTGTCACACTCCA3′) and of miRT1 (5′TTACATACTTCTTTACATTCCA3′) sequences cloned in the 3′ untranslated region of the expression cassette (Jimenez *et al*, 2013; Mallol *et al*, 2017). A non-coding cassette carrying the hAAT, the CAG, or the CMV promoter but no transgene was used to produce null vectors. Single-stranded AAV8 or AAV1 vectors were produced by triple transfection in HEK293 cells and purified using an optimized CsCl gradient-based purification protocol that renders vector preps of high purity and devoid of empty capsids (Ayuso *et al*, 2010). Viral genome titers were determined by quantitative PCR using linearized plasmid DNA as standard curve.

### Administration of AAV vectors

For systemic administration, AAV vectors were diluted in 200 μl of 0.001% F68 Pluronic® (Gibco) in PBS and injected via the tail vein. The intra-eWAT administration was performed as previously described (Jimenez *et al*, 2013). Briefly, mice were anesthetized with an intraperitoneal injection of ketamine (100 mg/kg) and xylazine (10 mg/kg). A laparotomy was performed in order to expose eWAT. AAV vectors were resuspended in PBS with 0.001% Pluronic® F68 (Gibco) and injected directly into the epididymal fat pad. Each epididymal fat pad was injected twice with 50 μl of the AAV solution. The abdomen was rinsed with sterile saline solution and closed with a two-layer suture. For intramuscular administration of AAV vectors, mice were anesthetized with an intraperitoneal injection of ketamine/xylazine. Hind limbs were shaved, and vectors were administered by intramuscular injection in a total volume of 180 μl

divided into six injection sites distributed between the quadriceps, gastrocnemius, and tibialis cranialis of each hind limb.

### Immunohistochemistry

Tissues were fixed for 12–24 h in 10% formalin, embedded in paraffin and sectioned. Sections were incubated overnight at 4°C with rat anti-Mac2 (1:50; CL8942AP; Cedarlane), guinea pig anti-insulin (1:100; I-8510; Sigma-Aldrich), or rabbit anti-glucagon (1:100; 219-01; Signet Labs). Biotinylated rabbit anti-rat (1:300; E0467; Dako), goat anti-rabbit IgG (Alexa Fluor 568-conjugated) (1:200; A11011; ThermoFisher), goat anti-guinea pig IgG (Alexa Fluor 488-conjugated) (1:300; A11073; ThermoFisher), or rabbit anti-guinea pig coupled to peroxidase (1:300; P0141; Dako) were used as secondary antibodies. The ABC peroxidase kit (Pierce) was used for immunodetection, and sections were counterstained in Mayer's hematoxylin. Hoechst (B2261; Sigma-Aldrich) was used for nuclear counterstaining of fluorescent specimens. PicroSirius Red staining and Masson's trichrome staining were used to evaluate fibrosis. Morphometric analysis of adipocyte size was performed in eWAT sections stained with hematoxylin–eosin as previously described (Muñoz et al, 2010). Four animals per group were used and at least 250 adipocytes/animal were analyzed. Images were obtained with a Nikon Eclipse 90i microscope (Nikon). The percentage of β-cell area in the pancreas was analyzed in two insulin-stained sections 200 μm apart, by dividing the area of all insulin+ cells in one section by the total pancreas area of that section. β-Cell mass was calculated by multiplying pancreas weight by percentage of β-cell area, as previously described (Jimenez et al, 2011).

### RNA analysis

Total RNA was obtained from different tissues using isolation reagent (Tripure, Roche, for liver and QIAzol, Qiagen, for adipose depots) and an RNeasy Minikit (Qiagen) and treated with DNAseI (Qiagen). One microgram of RNA was reverse-transcribed using the Transcriptor First Strand cDNA Synthesis kit (Roche). Real-time quantitative PCR (qRT–PCR) was performed in a Lightcycler (Roche) using the Lightcycler 480 SyBr Green I Master Mix (Roche) and the following primers: coFGF21: 5′CCTAACCAGGACGC CACAAG3′, 5′GTTCCACCATGCTCAGAGGG3′; F4/80: 5′CTTTGG CTATGGGCTTCCAGTC3′, 5′GCAAGGAGGACAGAGTTTATC3′; CD68: 5′GGGGCTCTTGGGAACTACAC3′, 5′CAAGCCCTCTTTAAGCCCCA3′; IL-1β: 5′TGTAATGAAAGACGGCACACC3′, 5′TCTTCTTTGGGTATTG CTTGG3′; TNF-α: 5′CATCTTCTCAAAATTCGAGTGACAA3′, 5′TGG GAGTAGACAAGGTACAACCC3′; Rplp0: 5′TCCCACCTTGTCTCCAG TCT3′, 5′ACTGGTCTAGGACCCGAGAAG3′; Phospho1: 5′AGCTGGA GACCAACAGTTTC3′, 5′TCCCTAGATAGGCATCGTAGT3′; Serca2b: 5′ACCTTTGCCGCTCATTTTCC3′, 5′AGGCTGCACACACTCTTTAC3′; RyR2: 5′ATGGCTTTAAGGCACAGCG3′, 5′CAGAGCCCGAATCATCC AGC3′; Collagen 1: 5′GACTGGAAGAGCGGAGAGTA3′, 5′CCTTGATG GCGTCCAGGTT3′; Glut1: 5′TCGGCCTCTTTGTTAATCGC3′, 5′TAAGC ACAGCAGCCACAAAG3′; Glut4: 5′TGGCCTTCTTTGAGATTGGC3′, 5′ACCCCATGCCGACAATGAAG3′; HKI: 5′ACGGTCAAAATGCTGCC TTC3′, 5′ATTCGTTCCTCCGAGATCCA3′; HKII: 5′TTGCTGAAGGAAG CCATTCG3′, 5′TGCTTCCAGTGCCAACAATG3′. Data were normalized to Rplp0 expression.

### Western blot analysis

iWAT and iBAT were homogenized in QIAzol Lysis Reagent (Qiagen), and the protein fraction was isolated from the organic phase following the manufacturer's instructions. Proteins were separated by 12% SDS–PAGE and analyzed by immunoblotting with rabbit polyclonal anti-UCP1 (1:1,000; ab10983; Abcam) and rabbit polyclonal anti-α-tubulin (1:1,000; ab4074; Abcam) antibodies. Detection was performed using ECL Plus detection reagent (Amersham Biosciences).

### Hormone and metabolite assays

Hepatic triglyceride and cholesterol content were determined by chloroform: methanol (2:1 vol/vol) extraction of total lipids, as described previously (Carr et al, 1993). Triglycerides and cholesterol were quantified spectrophotometrically using an enzymatic assay (Horiba-ABX) in a Pentra 400 Analyzer (Horiba-ABX). Glycemia was determined using a Glucometer Elite™ (Bayer) and insulin levels were measured using the Rat Insulin ELISA kit (90010, Crystal Chem). Glucagon levels were measured using a glucagon radioimmunoassay (#GL-32K, EMD Millipore). Serum FGF21, adiponectin, leptin, IGFBP1, and IGF1 were determined using the Mouse/Rat FGF-21 ELISA kit (MF2100, R&D Systems), the Mouse Adiponectin ELISA kit (80569, Crystal Chem), the Mouse Leptin ELISA kit (90030, Crystal Chem), the IGFBP1 (Mouse) ELISA kit (KA3054, Abnova), and the m/r IGF-I-ELISA kit (E25, Mediagnost), respectively.

### Insulin tolerance test

Insulin (Humulin Regular; Eli Lilly) was injected intraperitoneally at a dose of 0.75 IU/kg body weight to fed mice. Glycemia was measured in tail vein blood samples at the indicated time points.

### Glucose tolerance test

Awake mice were fasted overnight (16 h) and administered with an intraperitoneal injection of glucose (2 g/kg body weight). Glycemia was measured in tail vein blood samples at the indicated time points. Venous blood was collected from tail vein in tubes (Microvette® CB 300, SARSTEDT) at the same time points and immediately centrifuged to separate serum, which was used to measure insulin levels.

### Indirect calorimetry

An indirect open circuit calorimeter (Oxylet, Panlab) was used to monitor $O_2$ consumption and $CO_2$ production. Mice were individualized and acclimated to the metabolic chambers for 24 h, and data were collected in each cage for 3 min, every 15 min, for 24 h. Data were taken during the light and dark cycles and were adjusted by body weight.

### Bone analysis

Bone volume and architecture were evaluated by μCT. Mouse tibiae were fixed in neutral-buffered formalin (10%) and scanned using

**The paper explained**

**Problem**

As the prevalence of type 2 diabetes (T2D) and obesity increases worldwide, so does the need for better therapies suited for the heterogeneous obese/T2D patient population. Fibroblast growth factor 21 (FGF21) is a promising therapeutic agent for T2D/obesity, but the native FGF21 protein has poor pharmacokinetic properties, including a short half-life and susceptibility to *in vivo* proteolytic degradation, which have hampered the pharmacological development of FGF21-based drug products.

**Results**

In this study, we took advantage of adeno-associated viral vectors (AAV) to genetically engineer tissues and obtain sustained circulating levels of FGF21. Gene transfer to the liver of animals fed a high-fat diet for a long time or of ob/ob mice resulted in marked reductions in body weight, adiposity, non-alcoholic liver disease, and insulin resistance for > 1 year. Similar observations were made when FGF21 was overexpressed and secreted from epididymal white adipose tissue of ob/ob mice. Furthermore, FGF21 production following the engineering of skeletal muscle of healthy animals fed a standard diet prevented the increase in weight and insulin resistance associated with aging. Importantly, these therapeutic effects were obtained in the absence of side effects despite continuously elevated serum FGF21.

**Impact**

Our results show that FGF21 gene therapy holds great translational potential in the fight against insulin resistance, T2D, obesity, and related comorbidities.

the eXplore Locus CT scanner (General Electric) at 27-μm resolution. Trabeculae were analyzed in 1 mm³ of proximal tibial epiphysis and 1.8 mm³ of cortical tibial diaphysis in four mice/group. Bone parameters were calculated with the MicroView 3D Image Viewer & Analysis Tool. The length of the tibia was measured from the intercondylar eminence to the medial malleolus.

**Open-field test**

The open-field test was performed between 9:00 am and 1:00 pm as previously reported (Haurigot *et al*, 2013). Briefly, animals were placed in the center of a brightly lit chamber (41 × 41 × 30 cm) crossed by two bundles of photobeams (LE 8811; Panlab) that detect horizontal and vertical movements. Motor and exploratory activities were evaluated during the first 6 min. The total distance covered was evaluated using a video tracking system (SMART Junior; Panlab).

**Statistical analysis and data processing**

Sample size determination was based on previous experience with similar studies. Randomization was performed using the excel function Roundup() or by GraphPad QuickCalcs to allocate mice in each group. In addition, we tested that the mean body weight and the mean glycemia were statistically not different for each experimental group prior to assignment to diet and/or treatment groups. Furthermore, each experimental group was caged separately to avoid any caging effects. All tests (ITT, GTT, Open-field, etc.) were performed by investigators blinded to the treatment. All results are expressed as mean ± SEM. The GraphPad Prism 7 software was used for statistical analyses. Data were analyzed by one-way ANOVA with Tukey's post hoc correction, except for those parameters involving comparison of only two experimental groups, in which case an unpaired Student's *t*-test was used. Differences were considered significant when $P < 0.05$. All $P$-values for main figures, EV figures, and appendix figures can be found in Appendix Tables S1–S36.

**Expanded View** for this article is available online.

## Acknowledgements

This work was supported by grants from Ministerio de Economía y Competitividad (MINECO) and FEDER, Plan Nacional I+D+I (SAF2014-54866R), and Generalitat de Catalunya (2014 SGR 1669 and ICREA Academia Award to F.B.), Spain, from the European Commission (MYOCURE, PHC-14-2015-667751) and the European Foundation for the Study of Diabetes (EFSD/MSD European Research Programme on Novel Therapies for Type 2 Diabetes, 2013). V.J. was recipient of a post-doctoral research fellowship from EFSD/Lilly. E.C., V.S., and C.M. received a predoctoral fellowship from Ministerio de Educación, Cultura y Deporte, and J.R. from Ministerio de Economía y Competitividad, Spain. The authors thank Marta Moya and Maria Molas for technical assistance.

## Author contributions

VJ, CJ, and FB designed and supervised experiments, and wrote and edited the manuscript. VJ, CJ, EC, VS, SMu, SD, JRo, CM, MG, IG, and GE generated reagents and performed experiments. XL generated reagents. IE, AC, SMa, AR, TF, SMo, FM, MN, and JRu performed experiments. SF and JRu analyzed data and contributed to discussion. VJ, CJ, VH, and FB analyzed data, contributed to discussion, and reviewed/edited manuscript.

## Conflict of interest

Veronica Jimenez, Claudia Jambrina, and Fatima Bosch are co-inventors on a patent application for the use of AAV vectors for the treatment of metabolic disorders.

## For more information

(i)   European Association for the Study of Diabetes (EASD): https://www.easd.org/
(ii)  American Diabetes Association (ADA): www.diabetes.org
(iii) International Diabetes Federation (IDF) Diabetes Atlas: http://www.diabetesatlas.org/
(iv)  European Society of Gene and Cell Therapy (ESGCT): www.esgct.eu
(v)   American Society of Gene & Cell Therapy (ASGCT): www.asgct.org

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
