## [Review Process File · EMBO Molecular Medicine]

FGF21 Gene Therapy as Treatment for Obesity and Insulin Resistance

Veronica Jimenez, Claudia Jambrina, Estefania Casana, Victor Sacristan, Sergio Muñoz, Sara Darriba, Jordi Rodó, Cristina Mallol, Miquel Garcia, Xavier León, Sara Marcó, Albert Ribera, Ivet Elias, Alba Casellas, Ignasi Grass, Gemma Elias, Tura Ferré, Sandra Motas, Sylvie Franckhauser, Francisca Mulero, Marc Navarro, Virginia Haurigot, Jesus Ruberte and Fatima Bosch

Review timeline:	Submission date:	18 December 2017
	Editorial Decision:	06 February 2018
	Revision received:	04 May 2018
	Editorial Decision:	24 May 2018
	Revision received:	08 June 2018
	Accepted:	14 June 2018

Editor: Céline Carret

Transaction Report:

1st Editorial Decision

06 February 2018

Thank you for the submission of your manuscript to EMBO Molecular Medicine and apologies for the delay in getting back to you. The holiday season always delayed editorial processes. We have now heard back from the three referees whom we asked to evaluate your manuscript.

You will see from the comments pasted below, that all 3 referees find the paper interesting and a well-executed study. Ref 1 and 2 have minor comments (aiming at strengthening the data mainly) while referee 3 is more critical: this referee refers to the somehow limited conceptual advance and potential toxic effects in large animals or after 1-year of treatment that reduce the translational relevance of the findings. After our cross-commenting exercise however, referee 3 reconsidered her/his position given that authors would not only address referees 1 and 2 concerns but also stress the novelty and important aspect that is the absence of toxicity after 1 year, as well as thoroughly discuss the need to move into larger animals for properly assessing the long-term toxicity issue of a sustained FGF21 therapy.

We would therefore welcome the submission of a revised version within three months for further consideration and would like to encourage you to address all the criticisms raised as suggested to improve conclusiveness and clarity. Please note that EMBO Molecular Medicine strongly supports a single round of revision and that, as acceptance or rejection of the manuscript will depend on another round of review, your responses should be as complete as possible.

I look forward to receiving your revised manuscript.

***** Reviewer's comments *****

Referee #1 (Comments on Novelty/Model System for Author):

This is an excellent paper with appropriate experimental design using a very state-of-the-art approach of gene therapy in widely used models.

Referee #1 (Remarks for Author):

Jimenez et al describe using AAV-mediated FGF21 overexpression in hepatocytes, adipose tissue and skeletal muscle to modulate glucose homeostasis. The observed therapeutic benefits such as improvements in insulin resistance, steatosis, weight and adiposity were maintained for a considerable amount of time without alterations in bone biology. The paper is clear, straightforward and well written, providing an attractive gene therapy approach to target FGF21 levels in vivo.

This referee only has minor concerns and feels this manuscript will become suitable for publication after the authors respond to the following questions.

1. Previous studies have reported beneficial effects of FGF21 administration on islet cell biology. How is islet morphology and physiology impacted in AAV8-hAAT-FGF21 treated mice (HFD or ob/ob)? For example what happens to GSIS and islet architecture? Glucagon secretion? Some of these could be included.
 2. In regard to immunogenicity and genotoxicity associated concerns with the usage of viral-mediated gene therapies do these mice develop anti-transgene immunity? For example are Tregs, CD8+ and CD4+ populations similar between null and FGF21 treated groups? Do these mice present similar serum proteinograms?
 3. Previous clinical trials have reported benefits on cholesterol and alterations in blood pressure. Do FGF21-treated mice show improvements in cholesterol levels? And is blood pressure affected by high circulating FGF21 levels? (In HFD or ob/ob treated mice).
- A few additional data, if possible, to respond to the points mentioned above would strengthen the manuscript.
4. Authors start discussing the increase in the prevalence of T2D by citing a paper from 1991. Perhaps a more recent review would be appropriate.
 5. One suggestion is for the authors to add introductory lines describing how FGF21 is cleared from the circulation
 6. It is unclear which statistical test was used to compare the groups in the different figures.
 7. Please confirm alpha-tubulin blot in Figure 3E. First lane seems larger compared to the UCPI blot.

Referee #2 (Remarks for Author):

Jimenez and colleagues presented the use of gene therapy using AAV-FGF21 as an alternative approach for the treatment of obesity/T2D in genetic or diet-induced obesity mice models. The authors showed that a long-term overexpression of FGF21 in the liver significantly improved body weight, adipose tissue mass, and inflammation, as well as hepatic steatosis, inflammation, fibrosis,

and neoplasms. A whole body increase in energy expenditure and improvement of glucose levels and insulin sensitivity were also described. Moreover, they demonstrated that epididymal adipose tissue and skeletal muscle could also be alternative host tissues for FGF21 overexpression with similar beneficial effects.

The novelty of the manuscript is the demonstration that long-term (> 1 year) effects of FGF21 overexpression do not result in apparent side effects, such as bone homeostasis disequilibrium, that is so far, the major concern about FGF21 therapy. It clearly indicates the long-term safety of gene therapy using AAVFGF21 vectors, which bring excitement to this field. The weakness of this work is the absent demonstration regarding the mechanism by which the long-term FGF21 can improve inflammation, steatosis, fibrosis, and neoplasms. Despite the significant improvement in the last decade in our understanding about the therapeutic role of FGF21, yet numerous factors remain to be defined, and others are a source of debate. Thus, understanding the long-term FGF21 therapeutically effects could considerably improve our understanding and fill some gaps in this field.

In general, the manuscript is well written. The methods are adequate for the study proposal, and the figures are clearly presented. I would suggest a few points to strengthen the authors' conclusions further.

1. The authors claim that the increased energy expenditure in mice infected with AVV8-hAAT-FGF21 may reflect changes in thermogenesis due to both decreased lipid content and increase UCP1 expression in the BAT. However, it has been shown that FGF21 therapy improves whole-body energy homeostasis in UCP1KO mice (Vénient et al., 2015). In addition, studies have demonstrated the maintenance of FGF21 metabolic benefits even after the surgical excision of interscapular brown adipose tissue (Camporez et al., 2013; Bernardo et al., 2015). Is there any evidence in this study that supports the UCP1 or BAT-dependency increase in energy expenditure?

2. By histological analysis and UCP1 protein content of iWAT, the authors determined that the metabolic benefits of FGF21 overexpression are independent of the browning of subcutaneous adipose tissue (figure 3F). Nonetheless, two recent studies demonstrated alternative mechanisms by which the subcutaneous adipose tissue could improve energy homeostases such as creatine-driven cycle (Kazak et al., 2015) and Serca2b dependent calcium cycling (Ikeda et al., 2017). Could the long-term FGF21 therapy acts on the subcutaneous adipose tissue by alternative mechanisms and improves the whole-body energy homeostasis? The author should include the contribution of such alternative pathways to support their conclusion.

3. Previous work demonstrated that FGF21 treatment attenuates hepatic fibrogenesis through TGF- β /smad2/3 and NF- κ B signaling pathways (Xu et al., 2016). Could the referred signaling pathways be responsible for the improvements in fibrosis in this study?

4. The authors showed that FGF21 overexpression increases the mice physical activity determined by an open field test. However, previous studies using transgenic and physiological (ketogenic diet) models to increase circulating FGF21 levels demonstrated that FGF21 acts on the CNS and decreasing the physical activity of those mice (Bookout et al. 2013). The authors wish to explain the discrepancy between both studies. Could the higher physical activity in the FGF21 treated group be an independent effect of body weight decrease?

5. Previous studies demonstrated that under a fasting condition, the liver-derived FGF21 acts on the hypothalamic-pituitary-adrenal (HPA) axis, increasing the systemic corticosterone levels, thereby stimulating hepatic gluconeogenesis. In fact, FGF21 KO mice present severe hypoglycemia under fasting condition (Liang et al., 2014). Here the authors describe an improvement of glycemic levels in Ob/Ob mice (Fig 8G) under FGF21 therapy. Is this effect mediated through hepatic gluconeogenesis suppression in Ob/Ob mice?

6. During the ITT (figure 5C) the higher titer AAV8-hAAT-FGF21 injected group (5×10^{10}) present robust decrease of blood glucose levels throughout the experiment (peak response after 60 min with ~80% reduction blood glucose levels). What is the basal (pre-insulin injection) and peak blood glucose concentration (mg/dL) for this group?

Referee #3 (Remarks for Author):

The paper by Jimenez et al describes the therapeutic effect on obesity, liver steatosis and insulin resistance of a single administration of AAV encoding FGF-21 in relevant mouse models.

This is a detailed, very well designed and executed study from a group with long-standing experience in diabetes. The results are sound and support the efficacy of FGF21 gene therapy.

This reviewer has however two general concerns:

-the overall originality of the study is somewhat limited. Indeed similar effects have been described when using multiple FGF21 protein administrations. As the authors point out, the short half-life of the growth factor requires frequent administrations and they show that this can be overcome by a single delivery of a gene therapy vector. This principle is also well described in the literature where there is evidence up to clinical trials that a single administration of AAV8 targets liver which is converted in a factory for sustained systemic secretion of therapeutic proteins like clotting factors or lysosomal enzymes.

-the second aspect relates to the safety of the approach. In this regard, the strength of the approach which relies on long-term expression of FGF21 is also its weakness should an adverse event occur. Although the authors did not observe the side effects described with FGF21 protein delivery, such as bone loss, or tumors, in fact they even describe a protective effect from high-fat diet-induced cancer, one can not exclude these or other side effects when moving to larger animals or humans. In this regard, studies that investigate both long-term safety of the approach in non-human primates or the use of a system for pharmacological regulation of FGF21 expression in the context of gene therapy would be very helpful to both address the issue of safety of FGF21 gene therapy as well as to add a layer of control over potentially toxic FGF21 expression. Of course this reviewer understands that these experiments are beyond the scope of this report, which indeed represents a very well done proof-of-concept of the efficacy of the approach in mice. However, without them, the translational potential of the approach remains to be established, which reduces one of its major strengths especially in the absence of a striking originality.

1st Revision - authors' response

04 May 2018

- **Referee #1**

Comments on Novelty/Model System for Author

This is an excellent paper with appropriate experimental design using a very state-of-the-art approach of gene therapy in widely used models.

Remarks for Author

Jimenez et al describe using AAV-mediated FGF21 overexpression in hepatocytes, adipose tissue and skeletal muscle to modulate glucose homeostasis. The observed therapeutic benefits such as improvements in insulin resistance, steatosis, weight and adiposity were maintained for a considerable amount of time without alterations in bone biology. The paper is clear, straightforward and well written, providing an attractive gene therapy approach to target FGF21 levels in vivo.

This referee only has minor concerns and feels this manuscript will become suitable for publication after the authors respond to the following questions.

We thank Referee 1 for appreciating the quality and relevance of our work and for helping us improving our manuscript through his/her suggestions.

1. Previous studies have reported beneficial effects of FGF21 administration on islet cell biology. How is islet morphology and physiology impacted in AAV8-hAAT-FGF21 treated mice (HFD or ob/ob)? For example, what happens to GSIS and islet architecture? Glucagon secretion? Some of these could be included.

We thank the Referee for raising these important issues. Following the Referee's advice, we evaluated islet morphology through double immunostaining for insulin and glucagon of pancreatic sections from HFD-fed AAV8-hAAT-FGF21-treated mice. Representative images of islets showed

normal distribution of α and β cells in these animals, with localization of glucagon-expressing cells in the periphery of the islet and of insulin-expressing cells in the core. These new data are now included in the revised version of the manuscript (Results, page 12; new Fig 6F; Materials and Methods, page 30).

To further evaluate the impact of AAV-FGF21 treatment on islets, we also performed a morphometric analysis of the β -cell mass of the pancreases obtained from HFD-fed mice treated with AAV8-hAAT-FGF21 vectors when adults. This analysis revealed that while AAV8-null-treated mice developed islet hyperplasia as a consequence of HFD feeding, the β -cell mass of animals treated with AAV8-hAAT-FGF21 vectors (at the doses of 2×10^{10} and 5×10^{10} vg/mouse) was similar to that of control mice fed a chow diet. This observation is now included in the revised version of the manuscript (Results, page 12; new Fig 6D and E; Materials and Methods, pages 30 and 31).

To assess islet physiology, we treated a new cohort of HFD-fed young adult mice with either 1×10^{10} or 5×10^{10} vg/mouse of AAV8-hAAT-FGF21 vectors to evaluate *in vivo* glucose stimulated insulin secretion. An intraperitoneal glucose tolerance test (GTT) (2 g glucose/kg bw) was performed 2 months after AAV administration. HFD-fed animals injected with either null or FGF21-encoding vectors at a dose of 1×10^{10} vg/mouse were glucose intolerant and showed markedly increased circulating levels of insulin during the GTT. In contrast, animals treated with 5×10^{10} vg/mouse of AAV8-hAAT-FGF21 showed improved glucose clearance when compared to chow-fed control mice. Insulin levels were indistinguishable between these two experimental groups. Moreover, HFD-fed animals treated with AAV8-hAAT-FGF21 vectors showed decreased circulating glucagon levels compared with HFD-fed null-treated mice. Hence, islet physiology was indeed improved at this therapeutic dose of FGF21-expressing vectors. These new data are included in the revised version of the manuscript (Results, pages 11 and 12; new Fig 6C; new Fig 7C and D; Materials and Methods, pages 32-33).

2. In regard to immunogenicity and genotoxicity associated concerns with the usage of viral-mediated gene therapies, do these mice develop anti-transgene immunity? For example, are Tregs, CD8+ and CD4+ populations similar between null and FGF21 treated groups? Do these mice present similar serum proteinograms?

In all our studies using AAV-FGF21 vectors, the native FGF21 protein is produced by transduced cells. In the case of AAV8-hAAT-FGF21 vectors, in which the vector serotype and the promoter were chosen to direct expression of the transgene to the liver after intravascular delivery, we documented stable circulating levels of FGF21 for more than 1 year after vector administration, supporting the absence of anti-transgene immune responses in these animals. This observation agreed with previous reports demonstrating that gene transfer to hepatocytes promotes induction of immune tolerance to transgene products (Dobrzynski *et al*, 2004; Ziegler *et al*, 2004; Zhang *et al*, 2004; Cooper *et al*, 2009; Lu & Song, 2009; Breous *et al*, 2009; Somanathan *et al*, 2010; Mingozzi *et al*, 2003). We also recorded steady levels of circulating FGF21 in mice injected intra-eWAT with AAV8-CAG-FGF21-dmiRT or intramuscularly with AAV1-CMV-FGF21. Regardless of the route of administration, this lack of immune responses reflects the fact that the FGF21 encoded by all the AAV vectors used in this study is the native FGF21 protein, and that all the mice used for the experiments expressed endogenous FGF21, facilitating the recognition of the transgene product as own by the immune system. This is in clear contrast to the situation in which exogenous proteins are used to mimic FGF21 pharmacological properties. The unfavourable pharmacokinetic properties of native FGF21, with short half-life, and high susceptibility to *in vivo* proteolytic degradation and *in vitro* aggregation, has obliged the use of FGF21 analogues/mimetics that differ structurally from the native protein -or may even be completely unrelated to it. Indeed, humoral responses have been observed shortly after administration of FGF21 analogues/mimetics to non-human primates and humans (Adams *et al*, 2013; Gaich *et al*, 2013; Talukdar *et al*, 2016;).

Undeniably, one of the greatest advantages of gene therapy is that it allows for the use of native FGF21, whose short half-life is compensated by the continuous production of the protein by transduced organs. This point is now discussed in the new version of the manuscript (pages 25 and 26).

3. *Previous clinical trials have reported benefits on cholesterol and alterations in blood pressure. Do FGF21-treated mice show improvements in cholesterol levels? And is blood pressure affected by high circulating FGF21 levels? (In HFD or ob/ob treated mice).*

Following the Referee's suggestion, we measured cholesterol content in the liver of all the cohorts of obese AAV-FGF21-treated mice (HFD-fed and ob/ob treated with AAV8-hAAAT-FGF21, ob/ob mice treated with AAV8-CAG-FGF21-dmiRT). A marked reduction in liver cholesterol content was observed in all FGF21-treated animals. In addition, we measured total hepatic triglyceride content in ob/ob mice treated with AAV8-hAAAT-FGF21 or AAV8-CAG-FGF21-dmiRT. In both cases, we observed a marked reduction in triglyceride content in AAV-FGF21-treated mice, which was dependent on the dose of vector administered. All these data suggest that treatment with FGF21-encoding vectors markedly improves lipid metabolism. These new observations have been included in the revised version of the manuscript (Results, pages 11, 15 and 18; new Fig 4C; new Appendix FigS5H and I; new Appendix FigS7E and F; Materials and Methods, page 32).

Unfortunately, we were unable to evaluate blood pressure as we currently do not have the necessary equipment to measure this parameter in house.

A few additional data, if possible, to respond to the points mentioned above would strengthen the manuscript.

We hope we have appropriately addressed the Referee's comments.

4. *Authors start discussing the increase in the prevalence of T2D by citing a paper from 1991. Perhaps a more recent review would be appropriate.*

Following the Referee's advice, we have now cite a more recent publication to support our statement regarding T2D prevalence.

5. *One suggestion is for the authors to add introductory lines describing how FGF21 is cleared from the circulation.*

Following the Referee's suggestion, we have now included information describing the clearance of FGF21 from the circulation through kidney excretion in the Introduction section of the revised manuscript (page 4).

6. *It is unclear which statistical test was used to compare the groups in the different figures.*

We apologize for not clearly specifying the statistical tests applied in each experiment in the original submitted version of the manuscript. In the Material and Methods section of the revised manuscript, we have now clearly indicated that data were analyzed by one-way ANOVA with Tukey's post hoc correction, except for those parameters involving comparison of only two experimental groups, in which case an unpaired Student's *t*-test was used. We have also indicated that the GraphPad Prism 7 software was used for statistical analyses (page 33).

7. *Please confirm alpha-tubulin blot in Figure 3E. First lane seems larger compared to the UCPI blot.*

We confirm that the alpha-tubulin blot in Figure 3E is the one that corresponds to the UCPI blot showed above. We believe the greater size of the band in the alpha-tubulin blot when compared to the size of the corresponding band in the UCPI blot is a result of the effect of the electrical field on the blot. We have observed this phenomenon in other blots.

- Referee #2

Remarks for Author

Jimenez and colleagues presented the use of gene therapy using AAV-FGF21 as an alternative approach for the treatment of obesity/T2D in genetic or diet-induced obesity mice models. The

authors showed that a long-term overexpression of FGF21 in the liver significantly improved body weight, adipose tissue mass, and inflammation, as well as hepatic steatosis, inflammation, fibrosis, and neoplasms. A whole body increase in energy expenditure and improvement of glucose levels and insulin sensitivity were also described. Moreover, they demonstrated that epididymal adipose tissue and skeletal muscle could also be alternative host tissues for FGF21 overexpression with similar beneficial effects.

The novelty of the manuscript is the demonstration that long-term (> 1 year) effects of FGF21 overexpression do not result in apparent side effects, such as bone homeostasis disequilibrium, that is so far, the major concern about FGF21 therapy. It clearly indicates the long-term safety of gene therapy using AAVFGF21 vectors, which bring excitement to this field. The weakness of this work is the absent demonstration regarding the mechanism by which the long-term FGF21 can improve inflammation, steatosis, fibrosis, and neoplasms. Despite the significant improvement in the last decade in our understanding about the therapeutic role of FGF21, yet numerous factors remain to be defined, and others are a source of debate. Thus, understanding the long-term FGF21 therapeutically effects could considerably improve our understanding and fill some gaps in this field.

In general, the manuscript is well written. The methods are adequate for the study proposal, and the figures are clearly presented. I would suggest a few points to strengthen the authors' conclusions further.

We thank Referee 2 for highlighting the quality and novelty of our work.

1. The authors claim that the increased energy expenditure in mice infected with AVV8-hAAT-FGF21 may reflect changes in thermogenesis due to both decreased lipid content and increase UCP1 expression in the BAT. However, it has been shown that FGF21 therapy improves whole-body energy homeostasis in UCP1KO mice (Véniant *et al.*, 2015). In addition, studies have demonstrated the maintenance of FGF21 metabolic benefits even after the surgical excision of interscapular brown adipose tissue (Camporez *et al.*, 2013; Bernardo *et al.*, 2015). Is there any evidence in this study that supports the UCP1 or BAT-dependency increase in energy expenditure?

As indicated by the Referee, it has been described that treatment of UCP1 knockout mice with a long-acting FGF21 analogue increased energy expenditure (Véniant *et al.*, 2015). In contrast to the phenotype reported by Véniant *et al.*, another study has described that FGF21-mediated increase in metabolic rate is blunted in UCP1 null mice (Samms *et al.*, 2015).

As pointed out by the Referee, the effects of FGF21 administration to mice are maintained after ablation of interscapular BAT (iBAT) (Bernardo *et al.*, 2015; Camporez *et al.*, 2013; Emanuelli *et al.*, 2014). In these studies, only the iBAT pad was surgically excised, what may in turn result in increased compensatory thermogenic activity in the remaining BAT depots of the animal and/or increased appearance of thermogenic-competent beige adipocytes in subcutaneous WAT. In agreement with this possibility, an increase in UCP1 expression and protein content were detected in the inguinal WAT of mice treated with FGF21 upon surgical removal of iBAT, compared with the levels observed in sham-operated FGF21-treated animals (Bernardo *et al.*, 2015). Therefore, we believe that it cannot be ruled out that that FGF21 effects on energy expenditure are UCP1-dependent.

In our study, treatment with AAV-FGF21 resulted in enhanced thermogenesis and in marked increase in UCP1 protein content in iBAT. This agrees with previously published data in small animals treated with recombinant native FGF21 protein or with FGF21 analogues/mimetics which demonstrated increased UCP1 in BAT and enhanced energy expenditure as a consequence of the treatment (Samms *et al.*, 2015; Véniant *et al.*, 2015; Owen *et al.*, 2014; Douris *et al.*, 2015). Nevertheless, we acknowledge that UCP1-independent mechanisms may also contribute to the enhancement of thermogenesis mediated by treatment with AAV-FGF21 (*see below, answer to question 2*).

We have incorporated two paragraphs to the Discussion of the new version of the manuscript (pages 22 and 23) in which we discuss the contribution of UCP1-dependent and independent mechanisms to the observed increase in energy expenditure.

2. *By histological analysis and UCP1 protein content of iWAT, the authors determined that the metabolic benefits of FGF21 overexpression are independent of the browning of subcutaneous adipose tissue (figure 3F). Nonetheless, two recent studies demonstrated alternative mechanisms by which the subcutaneous adipose tissue could improve energy homeostases such as creatine-driven cycle (Kazak et al., 2015) and Serca2b dependent calcium cycling (Ikeda et al., 2017). Could the long-term FGF21 therapy acts on the subcutaneous adipose tissue by alternative mechanisms and improves the whole-body energy homeostasis? The author should include the contribution of such alternative pathways to support their conclusion.*

We thank the Referee for this important suggestion. Following his/her advice, the levels of expression of Phospho1, an enzyme involved in the creatine-driven substrate cycle, were measured by quantitative PCR in iWAT of HFD-fed mice treated with AAV8-hAAT-FGF21 vectors. Treatment with 5×10^{10} vg/mouse of the therapeutic vector led to higher levels of expression of Phospho1 in iWAT of AAV8-hAAT-FGF21-treated mice than in age-matched, chow- and HFD-fed control groups, suggesting that the activity of the creatine-driven cycle was probably increased as a result of FGF21 gene transfer.

Regarding the calcium cycling-dependent thermogenic mechanism, no differences in the expression levels of Serca2b were detected in the iWAT of animals treated with AAV8-hAAT-FGF21 vectors when compared with chow- or HFD-fed null-treated animals. On the other hand, the iWAT expression of Ryr2, another enzyme involved in the same cycle, was increased by HFD-feeding in both null- and AAV8-hAAT-FGF21-treated mice. Altogether, these results suggest that the calcium cycling-dependent thermogenic mechanism is not involved in the improvement of whole-body energy homeostasis observed after AAV-FGF21 treatment.

These new observations are included in the revised version of the manuscript (Results, page 10; new Fig 3G; new Appendix FigS3C; Discussion page 23; Materials and Methods, page 31).

3. *Previous work demonstrated that FGF21 treatment attenuates hepatic fibrogenesis through TGF- β /smad2/3 and NF- κ B signaling pathways (Xu et al., 2016). Could the referred signaling pathways be responsible for the improvements in fibrosis in this study?*

Following the Referee's suggestion, we analyzed by Western blot the protein content of TGF- β and Smad2/3 in liver extracts of HFD-fed mice treated as young adults with 5×10^{10} vg/mouse. However, no differences were observed when compared to HFD-fed null mice. Similar results were obtained when phosphorylated Smad2/3 was evaluated in hepatic nuclear extracts. Likewise, the protein content of *I κ B α* and phosphorylated *I κ B α* in liver extracts as well as that of NF- κ B in hepatic nuclear extracts of the same cohorts of mice was not altered. These results suggest that the TGF- β /smad2/3 and NF- κ B signaling pathways did not contribute to the improvement of hepatic fibrosis observed in HFD-fed mice treated long-term with AAV-FGF21.

Moreover, further proof of the reduction in liver fibrosis in HFD-fed animals treated with AAV8-hAAT-FGF21 vectors, was obtained through PicroSirius Red staining, which specifically labels collagen 1 fibers. Collagen fibers were not detected in AAV-FGF21-treated mice. qPCR quantification of collagen 1 mRNA confirmed the reduction in the levels of expression of collagen 1 in the livers of these animals. These new data are included in the revised version of the manuscript (Results, page 11; new Fig5A, B; Materials and Methods, pages 30 and 31).

4. *The authors showed that FGF21 overexpression increases the mice physical activity determined by an open field test. However, previous studies using transgenic and physiological (ketogenic diet) models to increase circulating FGF21 levels demonstrated that FGF21 acts on the CNS and decreasing the physical activity of those mice (Bookout et al. 2013). The authors wish to explain the discrepancy between both studies. Could the higher physical activity in the FGF21 treated group be an independent effect of body weight decrease?*

A decrease in physical activity and in basal core temperature have previously been observed in both fasted transgenic mice overexpressing murine FGF21 specifically in the liver under control of the *ApoE* promoter (*ApoE*-mFGF21) and in lean mice fed a ketogenic diet (KD) (Inagaki et al, 2007; Bookout et al, 2013). This led to the hypothesis that FGF21 induces torpor. These effects were, however, observed mostly in animals experiencing a physiological situation similar to fasting

(Kharitonov & DiMarchi, 2017). ApoE-mFGF21 transgenic mice are energy-deprived as a result of a lifelong increase in metabolic rate, and they are overtly lean and smaller in size (Inagaki *et al*, 2007, 2008). Another point to consider in the analysis of this behavioral discrepancy is that the levels of circulating FGF21 in ApoE-mFGF21 animals range from 650 to 1000 ng/ml (Bookout *et al*, 2013), a concentration 25-40-fold higher than that achieved in HFD-fed mice following gene transfer with the high dose of AAV8-hAAT-FGF21 vectors. KD feeding also induces a phenotype that is compatible with energy deprivation; KD-fed mice are leaner than chow-fed animals and exhibit profound changes in metabolism and energy homeostasis (Badman *et al*, 2007; Bookout *et al*, 2013).

Other studies have reported observations in agreement with our results and opposed to those reported by Inagaki *et al.* and Bookout *et al.* in ApoE-mFGF21 transgenic and KD-fed mice. In a different line of FGF21 transgenic mice, in which overexpression of human FGF21 was also driven by the *ApoE* promoter (ApoE-hFGF21), Kharitonov and colleagues observed no differences in body temperature when mice were fed a HFD (Kharitonov *et al*, 2005), which suggested that the effects of FGF21 may be dependent on nutrient context (Solon-Biet *et al*, 2016). Similar to HFD-fed ApoE-hFGF21 transgenic mice, treatment of HFD-fed WT mice with recombinant FGF21 also augmented body temperature and/or physical activity (Coskun *et al*, 2008; Xu *et al*, 2009a). In our study, intravascular administration of AAV8-hAAT-FGF21 vectors to HFD fed mice also resulted in increased locomotor activity. All these studies argue against the possibility of FGF21 being a torpor-promoting factor. Finally, an allele of human FGF21 has very recently been associated with lower physical activity in humans (Frayling *et al*, 2018). However, this allele is very likely to represent decreased FGF21 function (Frayling *et al*, 2018).

Whether the higher physical activity documented in AAV-FGF21-treated mice is a direct effect of the protein, or an indirect effect of the lower body weight -and consequential greater agility- of AAV-FGF21-treated animals, remains to be completely elucidated. Supporting an FGF21-mediated central modulation of locomotor activity in HFD-fed mice treated with AAV8-hAAT-FGF21 vectors, FGF21 has been described to be able cross the blood-brain barrier (Hsueh *et al*, 2007) and regulate metabolism, physical activity and circadian behaviour through direct actions on the CNS (Bookout *et al*, 2013).

We have modified the original version of the manuscript to include the discussion of the effects of FGF21 on physical activity (pages 21-22).

5. Previous studies demonstrated that under a fasting condition, the liver-derived FGF21 acts on the hypothalamic-pituitary-adrenal (HPA) axis, increasing the systemic corticosterone levels, thereby stimulating hepatic gluconeogenesis. In fact, FGF21 KO mice present severe hypoglycemia under fasting condition (Liang *et al.*, 2014). Here the authors describe an improvement of glycemic levels in Ob/Ob mice (Fig 8G) under FGF21 therapy. Is this effect mediated through hepatic gluconeogenesis suppression in Ob/Ob mice?

Following the Referee's suggestion, we evaluated whether the decrease in circulating glucose levels observed in ob/ob mice after AAV-FGF21 treatment resulted from suppression of hepatic gluconeogenesis by measuring the expression of phosphoenolpyruvate carboxykinase (PEPCK) and glucose-6-phosphatase (G6Pase) by qPCR. No changes in the expression of these enzymes were observed in the liver of ob/ob mice in which AAV-mediated FGF21 expression was targeted to the liver, except for the animals treated with 1×10^{11} vg of AAV8-hAAT-FGF21 that showed increased PEPCK expression (new Appendix FigS6A, B). These results suggested that AAV-mediated long-term expression of FGF21 in the liver, and the subsequent increase of circulating FGF21, did not lower glucose by inhibiting hepatic glucose production.

In support of our observations, a few reports have suggested that the glucose-lowering effects of FGF21 are independent of hepatic glucose production or improved hepatic insulin sensitivity (Xu *et al*, 2009b; Ding *et al*, 2012; Camacho *et al*, 2013; Emanuelli *et al*, 2014). Thus, insulin sensitization is considered to be the main mechanism by which FGF21 improves glycemic control (Berglund *et al*, 2009; Holland *et al*, 2013; Lin *et al*, 2013). It is noteworthy that in the present study we only detected amelioration of glycemia in those ob/ob mice treated with AAV8-FGF21 vectors that also showed improved insulin sensitivity (original Fig 7I and Fig 8I, now new Fig 9I and Fig 10I). Moreover, it has been described that FGF21 glucose-lowering effects may also

be due to increased glucose utilization in white and brown adipocytes and to increased energy expenditure (Xu *et al*, 2009; Ding *et al*, 2012; Camacho *et al*, 2013; Emanuelli *et al*, 2014; Kharitonov *et al*, 2005; Hondares *et al*, 2010; Samms *et al*, 2015), and UCP1 has been reported to be essential for full FGF21 glucose-lowering effects by enhancing glucose uptake in BAT and UCP1-dependent thermogenesis (BonDurant *et al*, 2017; Kwon *et al*, 2015; Véniant *et al*, 2015; Samms *et al*, 2015).

To further evaluate the molecular mechanisms underlying the AAV-FGF21-mediated glucose lowering effects in ob/ob mice, we assessed the expression of key components of the glucose uptake machinery in different pads of adipose tissue (iWAT, eWAT and iBAT): glucose transporters GLUT1 and GLUT4, the glucose phosphorylating enzymes HKI and HKII, as well as UCP1 in iBAT. In AAV8-hAAT-FGF21 treated ob/ob mice, the expression of GLUT1 was increased in iWAT and iBAT, and that of GLUT4 was increased in eWAT, iWAT and iBAT. HKI and HKII were upregulated only in iBAT. Moreover, UCP1 expression was increased in the iBAT of ob/ob mice treated with the high dose of AAV8-hAAT-FGF21 vectors (new Appendix FigS6C-G). Altogether, these results suggest that the long-term amelioration of glycemia observed in ob/ob mice following treatment with AAV-FGF21 vectors probably results from increased glucose uptake by white and brown adipocytes and enhanced thermogenesis in iBAT.

These new results have been included in the revised version of the manuscript (Results, pages 15 and 16; new Appendix FigS6; Materials and Methods, page 31).

6. During the ITT (figure 5C) the higher titer AAV8-hAAT-FGF21 injected group (5×10^{10}) present robust decrease of blood glucose levels throughout the experiment (peak response after 60 min with ~80% reduction blood glucose levels). What is the basal (pre-insulin injection) and peak blood glucose concentration (mg/dL) for this group?

During the ITT, the basal and peak (minute 60) blood glucose levels of the HFD-fed mice treated when young adults with 5×10^{10} vg/mouse of AAV8-hAAT-FGF21 vectors were 166.4 ± 5.7 mg/dL and 37.9 ± 5.2 mg/dL, respectively. This information has been included in the figure legend of the new Fig 7A (original Fig 5C) of the revised manuscript (page 53).

- Referee #3

Remarks for Author

The paper by Jimenez et al describes the therapeutic effect on obesity, liver steatosis and insulin resistance of a single administration of AAV encoding FGF-21 in relevant mouse models. This is a detailed, very well designed and executed study from a group with long-standing experience in diabetes. The results are sound and support the efficacy of FGF21 gene therapy.

This reviewer has however two general concerns:

- the overall originality of the study is somewhat limited. Indeed, similar effects have been described when using multiple FGF21 protein administrations. As the authors point out, the short half-life of the growth factor requires frequent administrations and they show that this can be overcome by a single delivery of a gene therapy vector. This principle is also well described in the literature where there is evidence up to clinical trials that a single administration of AAV8 targets liver which is converted in a factory for sustained systemic secretion of therapeutic proteins like clotting factors or lysosomal enzymes.

We thank Referee 3 for appreciating the quality of our work, whose importance and novelty we would like to highlight.

As pointed out by the Referee, there is a vast amount of literature reporting that the repeated administration of recombinant native FGF21 protein or FGF21 analogues/mimetics has anti-obesogenic and anti-diabetic effects not only in small and large animal models but also in humans (Zhang & Li, 2015; So & Leung, 2016). However, none of these previous studies have provided proof of long-lasting therapeutic efficacy or safety, being the longest follow-up reported to

date of 9 weeks (Stanislaus *et al*, 2017). Indeed, many of the findings of these studies have been controversial, likely due to their short duration. Our study represents the longest (>1 year) follow-up ever reported for an FGF21-based treatment and the first demonstration of long-term counteraction of obesity, NASH and insulin resistance upon a single administration of a FGF21 gene therapy. It is noteworthy that mice were about 1.5-years-old at the time of sacrifice. Following >14 months of HFD feeding, and in contrast to obese and highly insulin resistant 16.5-month-old AAV-null-treated “old mice”, the FGF21 “old mice” were lean and had a metabolic phenotype similar to that of healthy control mice. Furthermore, this metabolic improvement was achieved in the absence of side effects on bones, despite the fact that AAV-FGF21-treated mice had continuously elevated levels of FGF21 in serum for over a year. Moreover, the high incidence of hepatocarcinoma (HCC) observed in HFD-fed, AAV-null-treated “old mice” was blunted, and FGF21 “old mice” were free of liver tumors, which was in agreement with the absence of hepatic steatosis and fibrosis -key hallmarks of NASH- in these animals. Despite the low number of animals, these results were striking, and to the best of our knowledge, this is the first time in which the effects of FGF21 on the incidence of HCC after long-term HFD-feeding is reported. Taken together, these results support the long-term efficacy and safety of our gene therapy approach

Our study also constitutes the first demonstration that AAV-FGF21 gene therapy can be efficacious when applied to obese mice aged 10 months at the time of treatment, a model that resembles more closely the clinical setting for human obesity and insulin resistance. Despite the fact that mice were under HFD feeding, AAV-FGF21 gene therapy mediated sustained clinical benefit (>8 months), and at 16.5-month of age these “old mice” were similar to healthy mice. These observations demonstrated that AAV-FGF21 treatment can counteract the disease once clearly established. No previous data was available in the literature using FGF21 protein or analogs/mimetics in old obese mice. Therefore, this part of our study is also novel.

Another novel aspect of the study is that AAV-FGF21 gene therapy has demonstrated similar efficacy after genetic engineering of three different tissues (liver, adipose tissue and skeletal muscle) as source of circulating FGF21. This is important because these studies were designed considering a potential future clinical development of this strategy; the possibility of engineering more than one tissue could help in the translation of the approach. In this regard, we showed that AAV8-mediated gene transfer of FGF21 to both the liver and the white adipose tissue afforded equally effective reversal of disease. This demonstrated that adipose tissue could be an attractive alternative target organ, particularly for those patients who are not eligible for liver-directed gene transfer due to underlying hepatic disease, such as cirrhosis or liver cancer. Moreover, we found that the levels of AAV-derived FGF21 in circulation necessary to mediate therapeutic effects were very similar (around 25-50 ng/mL) independent of the transduced organ that acted as source of the protein. These levels are much lower than the peak concentration achieved in clinical trials after periodic administration of FGF21 analogues (Talukdar *et al*, 2016; Dong *et al*, 2015).

It is true that the gene therapy field has advanced considerably in the past few years, with multiple clinical trials for several genetic diseases presently underway. Although similar approaches have been used to deliver genes to specific tissues (e.g. the liver), each disease is different, and the outcome of one approach is no guarantee of efficacy in the treatment of a different disease (even when the same vector and route of administration are used). A whole body of preclinical data demonstrating efficacy and safety is required before moving the approach to patients. Our work is the first preclinical proof-of-concept study demonstrating long-term efficacy for FGF21 gene therapy in the treatment of obesity and insulin resistance. Further studies in adequate large animal models are needed in order to move FGF21 gene therapy to the clinic for the treatment of these highly prevalent diseases.

- the second aspect relates to the safety of the approach. In this regard, the strength of the approach which relies on long-term expression of FGF21 is also its weakness should an adverse event occur. Although the authors did not observe the side effects described with FGF21 protein delivery, such as bone loss, or tumors, in fact they even describe a protective effect from high-fat diet-induced cancer, one can not exclude these or other side effects when moving to larger animals or humans. In this regard, studies that investigate both long-term safety of the approach in non-human primates or the use of a system for pharmacological regulation of FGF21 expression in the context of gene therapy would be very helpful to both address the issue of safety of FGF21 gene therapy as well as to add a layer of control over potentially toxic FGF21 expression. Of course this reviewer

understands that these experiments are beyond the scope of this report, which indeed represents a very well done proof-of-concept of the efficacy of the approach in mice. However, without them, the translational potential of the approach remains to be established, which reduces one of its major strengths especially in the absence of a striking originality.

We agree with the Referee in that the long-term safety of continuous FGF21 production upon gene delivery could be a potential concern. Studies that investigate the long-term safety and efficacy of the approach in large animals, including non-human primates, are mandatory before moving AAV-FGF21-mediated gene therapy to the clinic. Nevertheless, the present study is the first preclinical proof-of-concept in different mouse models of obesity and insulin resistance of the efficacy and safety of FGF21 gene therapy, and as such it represents the first step in the development of FGF21 gene therapy. Proof-of concept studies in small animals help choose the most suitable target organ and route of administration, choose the best product candidate and establish minimally effective doses. Due to many constraints, including the experimental costs and the lower numbers of animals involved in the studies, this type of assessments is difficult to make in large animal models. Although our long-term observations in mice aged 16.5 months indicated the lack of adverse events after almost a lifetime of FGF21 production, we cannot exclude that adverse events may occur when translating the approach to large animals.

The importance of testing the safety of FGF21 gene therapy in suitable large animal models is now discussed in page 28 of the revised version of the manuscript.

To increase the safety of our gene therapy approaches, our group is working in the development of inducible promoters and other regulatory sequences that, once included in the AAV constructs, will help drive the expression of genes of interest in a safer way.

References

- Adams AC, Halstead CA, Hansen BC, Irizarry AR, Martin JA, Myers SR, Reynolds VL, Smith HW, Wroblewski VJ & Kharitonov A (2013) LY2405319, an Engineered FGF21 Variant, Improves the Metabolic Status of Diabetic Monkeys. *PLoS One* **8**: e65763
- Badman MK, Pissios P, Kennedy AR, Koukos G, Flier JS & Maratos-Flier E (2007) Hepatic Fibroblast Growth Factor 21 Is Regulated by PPARalpha and Is a Key Mediator of Hepatic Lipid Metabolism in Ketotic States. *Cell Metab.* **5**: 426–437
- Berglund ED, Li CY, Bina HA, Lynes SE, Michael MD, Shanafelt AB, Kharitonov A & Wasserman DH (2009) Fibroblast Growth Factor 21 Controls Glycemia via Regulation of Hepatic Glucose Flux and Insulin Sensitivity. *Endocrinology* **150**: 4084–4093
- Bernardo B, Lu M, Bandyopadhyay G, Li P, Zhou Y, Huang J, Levin N, Tomas EM, Calle RA, Erion DM, Rolph TP, Brenner M & Talukdar S (2015) FGF21 does not require interscapular brown adipose tissue and improves liver metabolic profile in animal models of obesity and insulin-resistance. *Sci. Rep.* **5**:
- BonDurant LD, Ameka M, Naber MC, Markan KR, Idiga SO, Acevedo MR, Walsh SA, Ornitz DM & Potthoff MJ (2017) FGF21 Regulates Metabolism Through Adipose-Dependent and -Independent Mechanisms. *Cell Metab.* **25**: 935–944.e4
- Bookout AL, De Groot MHM, Owen BM, Lee S, Gautron L, Lawrence HL, Ding X, Elmquist JK, Takahashi JS, Mangelsdorf DJ & Kliewer SA (2013) FGF21 regulates metabolism and circadian behavior by acting on the nervous system. *Nat. Med.* **19**: 1147–1152
- Breous E, Somanathan S, Vandenberghe LH & Wilson JM (2009) Hepatic regulatory T cells and Kupffer cells are crucial mediators of systemic T cell tolerance to antigens targeting murine liver. *Hepatology* **50**: 612–621
- Camacho RC, Zafian PT, Achanfuo-Yeboah J, Manibusan A & Berger JP (2013) Pegylated Fgf21 rapidly normalizes insulin-stimulated glucose utilization in diet-induced insulin resistant mice. *Eur. J. Pharmacol.* **715**: 41–45
- Camporez JPG, Jornayvaz FR, Petersen MC, Pesta D, Guigni BA, Serr J, Zhang D, Kahn M, Samuel VT, Jurczak MJ & Shulman GI (2013) Cellular Mechanisms by Which FGF21 Improves Insulin Sensitivity in Male Mice. *Endocrinology* **154**: 3099–3109
- Cooper M, Nayak S, Hoffman BE, Terhorst C, Cao O & Herzog RW (2009) Improved Induction of Immune Tolerance to Factor IX by Hepatic AAV-8 Gene Transfer. *Hum. Gene Ther.* **20**: 767–776
- Coskun T, Bina HA, Schneider MA, Dunbar JD, Hu CC, Chen Y, Moller DE & Kharitonov A

- (2008) Fibroblast growth factor 21 corrects obesity in mice. *Endocrinology* **149**: 6018–6027
- Ding X, Boney-Montoya J, Owen BM, Bookout AL, Coate KC, Mangelsdorf DJ & Klier SA (2012) β Klotho Is Required for Fibroblast Growth Factor 21 Effects on Growth and Metabolism. *Cell Metab.* **16**: 387–393
- Dobrzynski E, Mingozzi F, Liu YL, Bendo E, Cao O, Wang L & Herzog RW (2004) Induction of antigen-specific CD4⁺ T-cell anergy and deletion by in vivo viral gene transfer. *Blood* **104**: 969–977
- Dong JQ, Rossulek M, Somayaji VR, Baltrukonis D, Liang Y, Hudson K, Hernandez-Illas M & Calle RA (2015) Pharmacokinetics and pharmacodynamics of PF-05231023, a novel long-acting FGF21 mimetic, in a first-in-human study. *Br. J. Clin. Pharmacol.* **80**: 1051–1063
- Douris N, Stevanovic DM, Fisher Ffolliott M, Cisu TI, Chee MJ, Nguyen NL, Zarebidaki E, Adams AC, Kharitonkov A, Flier JS, Bartness TJ & Maratos-Flier E (2015) Central Fibroblast Growth Factor 21 Browns White Fat via Sympathetic Action in Male Mice. *Endocrinology* **156**: 2470–2481
- Emanuelli B, Vienberg SG, Smyth G, Cheng C, Stanford KI, Arumugam M, Michael MD, Adams AC, Kharitonkov A & Kahn CR (2014) Interplay between FGF21 and insulin action in the liver regulates metabolism. *J. Clin. Invest.* **124**: 515–27
- Frayling TM, Beaumont RN, Jones SE, Yaghootkar H, Tuke MA, Ruth KS, Casanova F, West BB, Locke J, Sharp S, Ji Y, Thompson W, Harrison J, Etheridge AS, Gallins PJ, Jima D, Wright F, Zhou Y, Innocenti F, Lindgren CM, Grarup N, Murray A, Freathy RM, Weedon MN, Tyrrel JI & Wood AR (2018) A Common Allele in FGF21 Associated with Sugar Intake Is Associated with Body Shape, Lower Total Body-Fat Percentage, and Higher Blood Pressure. *Cell Reports* **23**: 327–336
- Gaich G, Chien JY, Fu H, Glass LC, Deeg MA, Holland WL, Kharitonkov A, Bumol T, Schilke HK & Moller DE (2013) The Effects of LY2405319, an FGF21 Analog, in Obese Human Subjects with Type 2 Diabetes. *Cell Metab.* **18**: 333–340
- Holland WL, Adams AC, Brozinick JT, Bui HH, Miyauchi Y, Kusminski CM, Bauer SM, Wade M, Singhal E, Cheng CC, Volk K, Kuo MS, Gordillo R, Kharitonkov A & Scherer PE (2013) An FGF21-adiponectin-ceramide axis controls energy expenditure and insulin action in mice. *Cell Metab.* **17**: 790–797
- Hondares E, Rosell M, Gonzalez FJ, Giralt M, Iglesias R & Villarroya F (2010) Hepatic FGF21 expression is induced at birth via PPAR α in response to milk intake and contributes to thermogenic activation of neonatal brown fat. *Cell Metab.* **11**: 206–12
- Hsueh H, Pan W & Kastin AJ (2007) The fasting polypeptide FGF21 can enter brain from blood. *Peptides* **28**: 2382–6
- Inagaki T, Dutchak P, Zhao G, Ding X, Gautron L, Parameswara V, Li Y, Goetz R, Mohammadi M, Esser V, Elmquist JK, Gerard RD, Burgess SC, Hammer RE, Mangelsdorf DJ & Klier SA (2007) Endocrine Regulation of the Fasting Response by PPAR α -Mediated Induction of Fibroblast Growth Factor 21. *Cell Metab.* **5**: 415–425
- Inagaki T, Lin VY, Goetz R, Mohammadi M, Mangelsdorf DJ & Klier SA (2008) Inhibition of growth hormone signaling by the fasting-induced hormone FGF21. *Cell Metab.* **8**: 77–83
- Kharitonkov A & DiMarchi R (2017) Fibroblast growth factor 21 night watch: advances and uncertainties in the field. *J. Intern. Med.* **281**: 233–246
- Kharitonkov A, Shiyanova TL, Koester A, Ford AM, Micanovic R, Galbreath EJ, Sandusky GE, Hammond LJ, Moyers JS, Owens RA, Gromada J, Brozinick JT, Hawkins ED, Wroblewski VJ, Li D-S, Mehrbod F, Jaskunas SR & Shanafelt AB (2005a) FGF-21 as a novel metabolic regulator. *J. Clin. Invest.* **115**: 1627–1635
- Kharitonkov A, Shiyanova TL, Koester A, Ford AM, Micanovic R, Galbreath EJ, Sandusky GE, Hammond LJ, Moyers JS, Owens RA, Gromada J, Brozinick JT, Hawkins ED, Wroblewski VJ, Li D-S, Mehrbod F, Jaskunas SR & Shanafelt AB (2005b) FGF-21 as a novel metabolic regulator. *J. Clin. Invest.* **115**: 1627–35
- Kim AM, Somayaji VR, Dong JQ, Rolph TP, Weng Y, Chabot JR, Gropp KE, Talukdar S & Calle RA (2017) Once-weekly administration of a long-acting fibroblast growth factor 21 analogue modulates lipids, bone turnover markers, blood pressure and body weight differently in obese people with hypertriglyceridaemia and in non-human primates. *Diabetes, Obes. Metab.*
- Kwon MM, O'Dwyer SM, Baker RK, Covey SD & Kieffer TJ (2015) FGF21-Mediated Improvements in Glucose Clearance Require Uncoupling Protein 1. *Cell Rep.* **13**: 1521–1527
- Lin Z, Tian H, Lam KSL, Lin S, Hoo RCL, Konishi M, Itoh N, Wang Y, Bornstein SR, Xu A & Li X (2013) Adiponectin mediates the metabolic effects of FGF21 on glucose homeostasis and insulin sensitivity in mice. *Cell Metab.* **17**: 779–789

- Lu Y & Song S (2009) Distinct immune responses to transgene products from rAAV1 and rAAV8 vectors. *Proc. Natl Acad. Sci. USA* **106**: 17158–62
- Mingozzi F, Liu Y-L, Dobrzynski E, Kaufhold A, Liu JH, Wang Y, Arruda VR, High KA & Herzog RW (2003) Induction of immune tolerance to coagulation factor IX antigen by in vivo hepatic gene transfer. *J. Clin. Invest.* **111**: 1347–56
- Owen BM, Ding X, Morgan DA, Coate KC, Bookout AL, Rahmouni K, Kliewer SA & Mangelsdorf DJ (2014) FGF21 acts centrally to induce sympathetic nerve activity, energy expenditure, and weight loss. *Cell Metab.* **20**: 670–677
- Samms RJ, Smith DP, Cheng CC, Antonellis PP, Perfield JW, Kharitonov A, Gimeno RE & Adams AC (2015a) Discrete Aspects of FGF21 In Vivo Pharmacology Do Not Require UCP1. *Cell Rep.* **11**: 991–999
- Samms RJ, Smith DP, Cheng CC, Antonellis PP, Perfield JW, Kharitonov A, Gimeno RE & Adams AC (2015b) Discrete Aspects of FGF21 In Vivo Pharmacology Do Not Require UCP1. *Cell Rep.* **11**: 991–9
- So WY & Leung PS (2016) Fibroblast Growth Factor 21 As an Emerging Therapeutic Target for Type 2 Diabetes Mellitus. *Med. Res. Rev.* **36**: 672–704
- Solon-Biet SM, Cogger VC, Pulpitel T, Heblinski M, Wahl D, McMahon AC, Warren A, Durrant-Whyte J, Walters KA, Krycer JR, Ponton F, Gokarn R, Wali JA, Ruohonen K, Conigrave AD, James DE, Raubenheimer D, Morrison CD, Le Couteur DG & Simpson SJ (2016) Defining the Nutritional and Metabolic Context of FGF21 Using the Geometric Framework. *Cell Metab.* **24**: 555–565
- Somanathan S, Breous E, Bell P & Wilson JM (2010) AAV vectors avoid inflammatory signals necessary to render transduced hepatocyte targets for destructive t cells. *Mol. Ther.* **18**: 977–982
- Stanislaus S, Hecht R, Yie J, Hager T, Hall M, Spahr C, Wang W, Weiszmann J, Li Y, Deng L, Winters D, Smith S, Zhou L, Li Y, Véniant MM & Xu J (2017) A Novel Fc-FGF21 With Improved Resistance to Proteolysis, Increased Affinity Toward β -Klotho, and Enhanced Efficacy in Mice and Cynomolgus Monkeys. *Endocrinology* **158**: 1314–1327
- Talukdar S, Zhou Y, Li D, Rossulek M, Dong J, Somayaji V, Weng Y, Clark R, Lanba A, Owen BM, Brenner MB, Trimmer JK, Groppe KE, Chabot JR, Erion DM, Rolph TP, Goodwin B & Calle RA (2016) A Long-Acting FGF21 Molecule, PF-05231023, Decreases Body Weight and Improves Lipid Profile in Non-human Primates and Type 2 Diabetic Subjects. *Cell Metab.* **23**: 427–440
- Véniant MM, Sivits G, Helmering J, Komorowski R, Lee J, Fan W, Moyer C & Lloyd DJ (2015) Pharmacologic Effects of FGF21 Are Independent of the “Browning” of White Adipose Tissue. *Cell Metab.* **21**: 731–8
- Xu J, Lloyd DJ, Hale C, Stanislaus S, Chen M, Sivits G, Vonderfecht S, Hecht R, Li Y-S, Lindberg RA, Chen J-L, Jung DY, Zhang Z, Ko H-J, Kim JK & Véniant MM (2009a) Fibroblast growth factor 21 reverses hepatic steatosis, increases energy expenditure, and improves insulin sensitivity in diet-induced obese mice. *Diabetes* **58**: 250–9
- Xu J, Stanislaus S, Chinookoswong N, Lau YY, Hager T, Patel J, Ge H, Weiszmann J, Lu S-C, Graham M, Busby J, Hecht R, Li Y-S, Li Y, Lindberg R & Veniant MM (2009b) Acute glucose-lowering and insulin-sensitizing action of FGF21 in insulin-resistant mouse models--association with liver and adipose tissue effects. *AJP Endocrinol. Metab.* **297**: E1105–E1114
- Zhang J & Li Y (2015) Fibroblast Growth Factor 21 Analogs for Treating Metabolic Disorders. *Front. Endocrinol. (Lausanne)*. **6**: 168
- Zhang J, Xu L, Haskins ME & Ponder KP (2004) Neonatal gene transfer with a retroviral vector results in tolerance to human factor IX in mice and dogs. *Blood* **103**: 143–151
- Ziegler RJ, Lonning SM, Armentano D, Li C, Souza DW, Cherry M, Ford C, Barbon CM, Desnick RJ, Gao G, Wilson JM, Peluso R, Godwin S, Carter BJ, Gregory RJ, Wadsworth SC & Cheng SH (2004) AAV2 vector harboring a liver-restricted promoter facilitates sustained expression of therapeutic levels of α -galactosidase A and the induction of immune tolerance in Fabry mice. *Mol. Ther.* **9**: 231–240

Thank you for the submission of your revised manuscript to EMBO Molecular Medicine. We have now received the enclosed reports from the referees that were asked to re-assess it. As you will see

the reviewers are now supportive and I am pleased to inform you that we will be able to accept your manuscript pending editorial final amendments.

Please submit your revised manuscript within two weeks. I look forward to seeing a revised form of your manuscript as soon as possible.

***** Reviewer's comments *****

Referee #1 (Remarks for Author):

The authors have carefully responded to all my critiques with additional data.

Referee #2 (Remarks for Author):

The authors addressed the reviewer's comment satisfactory.

Referee #3 (Remarks for Author):

The authors have discussed in the cover letter aspects relevant to the novelty of their findings, on some of which I agree. As also highlighted by another reviewer, their long-term safety data remain the most prominent finding of their report, therefore I still think that studies in large animals would be important in view of potential further translation, yet this could be deferred to a future publication.

2nd Revision - authors' response

08 June 2018

Authors made requested editorial changes.

Corresponding Author Name: Fatima Bosch
 Journal Submitted to: EMBO Molecular Medicine
 Manuscript Number: EMM-2017-08791